# Sparse but Wrong: Incorrect L0 Leads to Incorrect Features in Sparse Autoencoders

## Abstract

Sparse Autoencoders (SAEs) extract features from LLM internal activations, meant to correspond to interpretable concepts. A core SAE training hyperparameter is L0: how many SAE features should fire per token on average. Existing work compares SAE algorithms using sparsity–reconstruction tradeoff plots, implying L0 is a free parameter with no single correct value aside from its effect on reconstruction. In this work we study the effect of L0 on SAEs, and show that if L0 is not set correctly, the SAE fails to disentangle the underlying features of the LLM. If L0 is too low, the SAE will mix correlated features to improve reconstruction. If L0 is too high, the SAE finds degenerate solutions that also mix features. Further, we present a proxy metric that can help guide the search for the correct L0 for an SAE on a given training distribution. We show that our method finds the correct L0 in toy models and coincides with peak sparse probing performance in LLM SAEs. We find that most commonly used SAEs have an L0 that is too low. Our work shows that L0 must be set correctly to train SAEs with correct features.

## 1 Introduction

It is theorized that Large Language Models (LLMs) represent concepts as linear directions in representation space, known as the Linear Representation Hypothesis (LRH) (Elhage et al., 2022; Park et al., 2024). These concepts are nearly orthogonal linear directions, allowing the LLM to represent many more concepts than there are neurons, a phenomenon known as superposition (Elhage et al., 2022). However, superposition poses a challenge for interpretability, as neurons in the LLM are polysemantic, firing on many different concepts.

Sparse autoencoders (SAEs) are meant to reverse superposition, and extract interpretable, monosemantic latent features (Cunningham et al., 2024; Bricken et al., 2023) using sparse dictionary learning (Olshausen & Field, 1997). SAEs have the advantage of being unsupervised, and can be scaled to millions of neurons in its hidden layer (hereafter called "latents"[1]). When training an SAE, practitioners must decide on the sparsity of SAE, measured in terms of L0, or how many latents activate on average for a given input. [2] L0 is typically considered a neutral design choice: most of the literature evaluates SAEs at a range of L0 values, referring to this as a "sparsity–reconstruction tradeoff" (Gao et al., 2024; Rajamanoharan et al., 2024). While most practitioners would expect that too high an L0 will break the SAE (afterall, it is called a *sparse* autoencoder), the implication of "sparsity–reconstruction tradeoff" plots is that any sufficiently low L0 is equally valid.

However, recent work shows the same trend: low L0 SAEs perform worse on downstream tasks Kantamneni et al. (2025); Bussmann et al. (2025). What causes this degraded performance at low L0? In this work, we explore the effect of L0 on SAEs. We begin with toy model experiments using synthetic data, and show that if the L0 is too low, the SAE can "cheat" by mixing together components of correlated features, achieving better reconstruction compared to an SAE with correct, disentangled features. We consider this to be a manifestation of feature hedging (Chanin et al.,

---

[1]We use *latents* to prevent overloading the term *feature*, which we reserve for human-interpretable concepts the SAE may capture. This breaks from earlier usage which used *feature* for both (Elhage et al., 2022), but aligns with the terminology in (Lieberum et al., 2024) and makes the distinction more clear.

[2]TopK and BatchTopK SAEs (Gao et al., 2024; Bussmann et al., 2024) set the L0 ($K$) directly, whereas L1 and JumpReLU (Cunningham et al., 2024; Bricken et al., 2023; Rajamanoharan et al., 2024) adjust it via a coefficient in the loss. In any case, all SAE trainers must decide on the target L0.

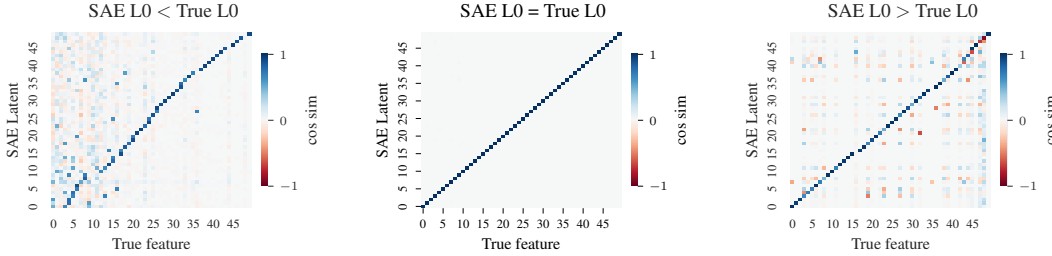

Figure 1: When SAE L0 is too low (left) or too high (right), the SAE mixes together correlated features, ruining monsemanticity. Only at the correct L0 (middle), the SAE learns correct features.

2025), where the SAE abuses feature correlations to compensate for insufficient resources to model the underlying features monsemantically. This mixing of correlated features into SAE latents affects both positively and negatively correlated features, meaning that in low L0 SAEs, nearly all latents are both less interpretable and more noisy than an SAE with a correctly set L0.

Our findings also show that "sparsity–reconstruction tradeoff" plots, commonly used to assess SAE architectures, are not a sound method of evaluating SAEs. We demonstrate using toy model experiments that at low L0, an SAE with ground-truth correct latents achieves worse reconstruction than an SAE that mixes correlated features. Thus, if we had an SAE training method that resulted in perfect SAEs, "sparsity–reconstruction tradeoff" plots would cause us to reject that method.

Finally, we develop a proxy metric based on projections between the SAE decoder and training activations that can detect if L0 is too low. We validate these findings on Gemma-2-2b (Team et al., 2024), demonstrating that decoder patterns similar to what we observe in our toy model experiments also manifests in LLM SAEs. We further validate that the optimal L0 we find with our method in Gemma-2-2b matches peak performance on sparse probing tasks (Kantamneni et al., 2025).

Our findings are of direct importance to anyone using SAEs in practice, showing that L0 must be set correctly for SAEs to learn correct features. Furthermore, our work implies that most SAEs used by researchers today have too low an L0.

## 2 BACKGROUND

**Sparse autoencoders (SAEs).** An SAE decomposes an input activation $\mathbf{x} \in \mathbb{R}^d$ into a hidden state, $\mathbf{a}$, consisting of $h$ hidden neurons, called "latents". An SAE is composed of an encoder $\mathbf{W}_{\text{enc}} \in \mathbb{R}^{h \times d}$, a decoder $\mathbf{W}_{\text{dec}} \in \mathbb{R}^{d \times h}$, a decoder bias $\mathbf{b}_{\text{dec}} \in \mathbb{R}^d$, and encoder bias $\mathbf{b}_{\text{enc}} \in \mathbb{R}^h$, and a nonlinearity $\sigma$, typically ReLU or a variant like JumpReLU (Rajamanoharan et al., 2024), TopK (Gao et al., 2024) or BatchTopK (Bussmann et al., 2024). The decoder is sometimes called the *dictionary*, in reference to sparse dictionary learning. We use both terms interchangeably.

$$\mathbf{a} = \sigma(\mathbf{W}_{\text{enc}}(\mathbf{x} - \mathbf{b}_{\text{dec}}) + \mathbf{b}_{\text{enc}}) \tag{1}$$
$$\hat{\mathbf{x}} = \mathbf{W}_{\text{dec}}\mathbf{a} + \mathbf{b}_{\text{dec}} \tag{2}$$

In this work we focus on BatchTopK and JumpReLU SAEs as these are both considered SOTA architectures. The JumpReLU activation is a modified ReLU with a threshold parameter $\tau > 0$, so $\text{JumpReLU}_\tau(x) = x \cdot \mathbf{1}_{x>\tau}$. The BatchTopK activation function selects the top $b \times k$ activations across a batch of size $b$, allowing variance in the $k$ selected per sample in the batch. After training, a BatchTopK SAE is converted to a JumpReLU SAE with a global $\tau$. We follow the JumpReLU training procedure outlined by Anthropic (Conerly et al., 2025).

SAEs are trained as follows, with an auxiliary loss $\mathcal{L}_p$ to revive dead latents with corresponding coefficient $\lambda_p$. JumpReLU SAEs also have a sparsity loss $\mathcal{L}_s$ and corresponding coefficient $\lambda_s$.

$$\mathcal{L} = \|\mathbf{x} - \hat{\mathbf{x}}\|_2^2 + \lambda_s \mathcal{L}_s + \lambda_p \mathcal{L}_p \tag{3}$$

The formulation of $\mathcal{L}_s$ and $\mathcal{L}_p$ for JumpReLU and BatchTopK SAEs is shown in Appendix A.1.

## 3  TOY MODEL EXPERIMENTS

The Linear Representation Hypothesis (LRH) (Elhage et al., 2022; Park et al., 2024) states that LLMs represent concepts (alternatively referred to as "features") as (nearly) orthogonal linear directions in representation space. Thus, the hidden activations in an LLM are simply the sum all the firing feature vectors (a feature direction with a positive, non-zero magnitude) that are being represented. While an LLM can represent a potentially large number of concepts this way, in any given activation, only a small number of concepts are actively represented.

For instance, if we inspect a hidden activation from within an LLM at the token "␣Canada", we may expect this activation to be a sum of feature vectors representing concepts like "country", "North America", "starts with C", "noun", etc... The job of a sparse autoencoder is to recover these "true feature" directions in its dictionary.

In a real LLM, we do not have ground-truth knowledge of the "true features" the model is representing, so we do not know if the SAE has learned the correct features. Fortunately, it is easy to create a toy model setup that follows the requirements of the LRH while providing ground-truth knowledge of the underlying true features.

Our toy model has a set of feature embeddings $\mathbf{F} \in \mathbb{R}^{g \times d}$, where $d$ is the input dimension of our SAE, and $g$ is the number of features. All features are orthogonal, so $f_i \cdot f_j = 0$ for $i \neq j$. Each feature $f_i$ fires with probability $p_i$, mean magnitude $\mu_i$, and magnitude standard deviation $\sigma_i$. Feature activations follow a correlated Bernoulli process controlled by correlation matrix $C$, with final magnitudes given by $m_i = a_i \cdot \text{ReLU}(\mu_i + \sigma_i \epsilon_i)$, where $a_i$ indicates whether feature $i$ is active and $\epsilon_i \sim \mathcal{N}(0, 1)$. Training activations for an SAE, $x \in \mathbb{R}^d$, are thus generated as $x = \sum_{i=1}^{n} m_i f_i$

In these toy model experiments, we mainly focus on BatchTopK SAEs (Bussmann et al., 2024) as this enables direct control of L0. Additionally, we validate our results with JumpReLU SAEs. We train SAEs on 15M synthetic samples with batch size 500 using SAELens (Bloom et al., 2024).

Throughout this section we will use the following terminology:

**True L0**  In toy models we have complete control over which features fire, so we know how many features are firing on average. We refer to this as the *true L0* of the toy model.

**Ground-truth SAE**  Since we know the ground-truth features in our toy models, we can construct an SAE that perfectly captures these features. We refer to this as the *ground-truth SAE*. This is an SAE where $g = h$, $\mathbf{W}_{\text{enc}} = \mathbf{F}^T$, $\mathbf{W}_{\text{dec}} = \mathbf{F}$, $\mathbf{b}_{\text{enc}} = 0$, $\mathbf{b}_{\text{dec}} = 0$.

### 3.1  LOW L0 SAES MIX CORRELATED AND ANTI-CORRELATED FEATURES

We begin with a small toy model with 5 true features ($g = 5$) in an input space of $d = 20$. We set each $p_i = 0.4$ such that on average 2 features are active per input, for a true L0 of 2. We begin with a simple correlation pattern between features, where $f_0$ is positively correlated with every feature $f_1$ through $f_4$, but otherwise there are no other correlations. We then train an SAE with $L0 = 2$, matching the true L0 of the model, and an SAE with slightly lower value of $L0 = 1.8$ (BatchTopK SAEs permit setting fractional L0). For the $L0 = 1.8$ SAE, we initialize it to the ground-truth solution, ensuring that the result of training is due to gradient pressure rather than just being a local minimum. We show the toy model feature correlation matrix as well as decoder cosine similarity plots with the true features for both SAEs in Figure 2.

When the SAE L0 matches the true L0, we see that the SAE perfectly learns the underlying true features. However, when SAE L0 is smaller than the true L0, the resulting SAE latents mix feature components together based on the correlation matrix. The latents tracking features $f_1$ through $f_4$ all mix in a *positive* component of $f_0$, but they have no components of each other.

Next, we invert the correlation, i.e. each feature $f_1$ through $f_4$ is negatively correlated with $f_0$ instead, while keeping everything else unchanged. We show the correlation matrix and SAE decoder cosine similarity with true features plots in Figure 3.

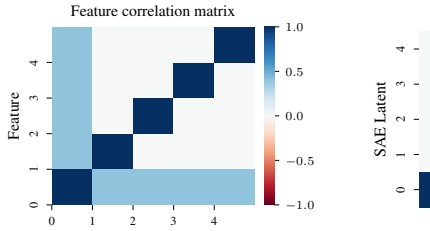 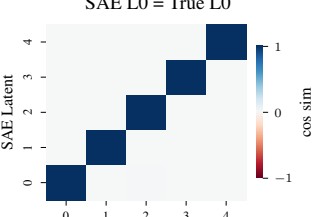 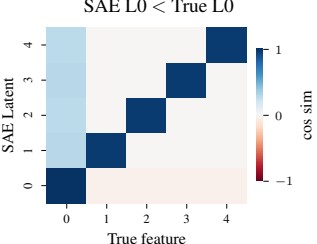

Figure 2: (left) Toy model feature correlation matrix showing positive correlations between features. (middle) SAE decoder cosine similarities with true feature when SAE L0 = 2, matching the true L0 of the toy model. (right) SAE decoder cosine similarities with true features when SAE L0 = 1.8. When L0 is too low, the SAE mixes components of features based on their firing correlations.

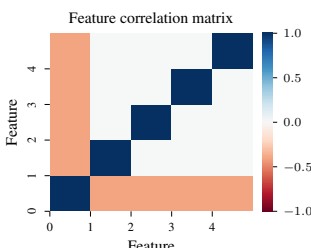 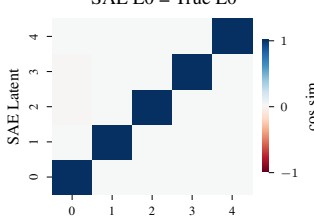 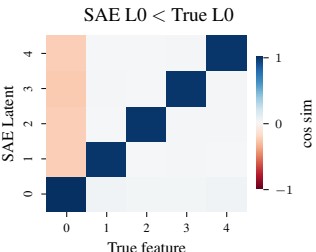

Figure 3: (left) Toy model feature correlation matrix showing negative correlations between features. (middle) SAE decoder cosine similarities with true feature when SAE L0 = 2, matching the true L0 of the toy model. (right) SAE decoder cosine similarities with true features when SAE L0 = 1.8,. When L0 is too low, the SAE mixes negative components of anti-correlated features.

Now, we see the same pattern as with positive correlations except inverted. The latents tracking features $f_1$ through $f_4$ mix in a *negative* component of $f_0$, but have no component of each other.

This pattern is problematic because it means that if our L0 is too low, every SAE latent will contain positive components of every positively correlated feature, and negative components of every negatively correlated feature in the model. Negative correlations are particularly bad, as negative correlations are prevalent throughout language. For instance, we may expect a nonsensical negative component of "Harry Potter" to appear in the latent for "French poetry", since Harry Potter has nothing to do with French poetry. This will result in highly polysemantic and noisy SAE latents.

Extended toy model experiments are shown in Appendix A.3.

## 3.2 LARGER TOY MODEL EXPERIMENTS

Next, we scale up to a larger toy model with 50 true features ($g = 50$) in input space of $d = 100$. We set $p_0 = 0.345$ and linearly decrease to $p_{49} = 0.05$, so firing probability decreases with feature number. The true L0 of this model is 11. We randomly generate a correlation matrix, so the firings of each feature are correlated with other features. Feature correlations are shown in Appendix A.2.

We train SAEs with L0 values that are too small ($L0 = 5$), exactly correct ($L0 = 11$), and too large ($L0 = 18$). Results are shown in Figure 1. When the SAE L0 matches the true L0, the SAE exactly learns the true features. When SAE L0 is too low, the SAE mixes components of correlated features together, particularly breaking latents tracking high-frequency features. When L0 is too high, the SAE learns degenerate solutions that mix features together. The further SAE L0 is from the true L0, the worse the SAE. Interestingly, when L0 is too high the SAE still learns many correct latents, but *when L0 is too low, every latent in the SAE is affected*.

### 3.3 MSE LOSS INCENTIVIZES LOW-L0 SAES TO MIX CORRELATED FEATURES

Why do SAEs with low L0 not learn the true features? We construct a ground-truth SAE and set $L0 = 5$, to match the low L0 SAE from Figure 1. We generate 100k synthetic training samples and calculate the Mean Square Error (MSE) of both these SAEs. The trained SAE with incorrect latents achieves a MSE of $2.73$, while the ground-truth SAE achieves a much worse MSE of $4.88$. Thus, *MSE loss actively incentivizes low L0 SAEs to learn incorrect latents.*

### 3.4 THE SPARSITY–RECONSTRUCTION TRADEOFF

It is common practice to evaluate SAE architectures using a sparsity–reconstruction tradeoff plot (Cunningham et al., 2024; Gao et al., 2024; Rajamanoharan et al., 2024), where the assumption is that having better reconstruction at a given sparsity is inherently better, and indicates that the SAE is correct. Afterall, we train SAEs to reconstruct inputs, so surely an SAE that has better reconstruction must therefore be a better SAE than one that has lower reconstruction?

Sadly, this is not the case. As we discussed in Section 3.3, when the L0 of the SAE is lower than optimal, the SAE can find ways to "cheat" by engaging in feature hedging (Chanin et al., 2025), and get a better MSE score by mixing components of correlated features together. This results in an SAE where the latents are not monosemantic, and do not track ground-truth features.

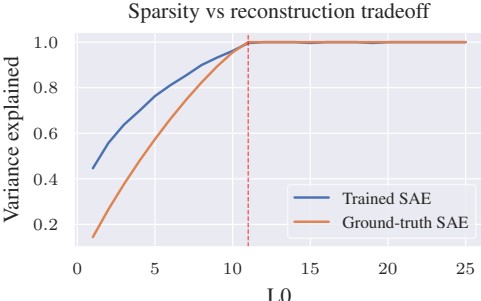

Figure 4: Sparsity ($L0$, lower is better) vs reconstruction (variance explained, higher is better) for learned SAEs and a ground-truth SAE. When L0 is less than the true L0 of the toy model (the dotted line), the trained SAE gets better reconstruction than the ground-truth SAE. Sparsity–reconstruction plots like this lead us to the incorrect conclusion that the ground-truth SAE is a worse SAE.

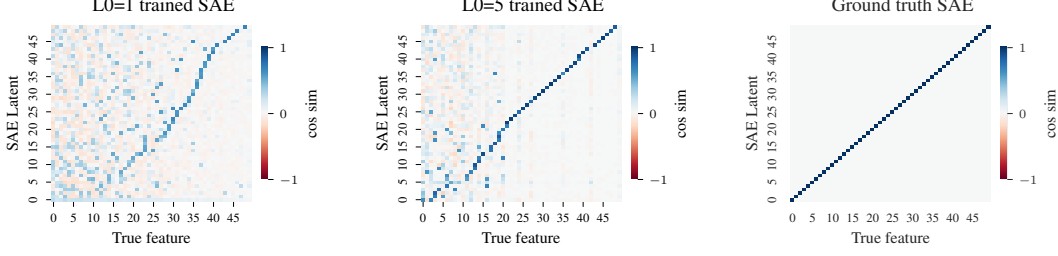

Figure 5: SAE decoder cosine similarity with true features for the SAEs from Figure 4 with L0=1 (left) and L0=5 (middle), compared with the ground-truth SAE (right). The trained SAEs score much better than the ground truth SAE on variance explained, despite their corrupted, polysemantic latents.

We next explore the sparsity–reconstruction tradeoff by training SAEs on our toy model at various L0s. Since we know the ground-truth features in our toy model, we construct a ground-truth SAE that perfectly represents these features. We vary the L0 of the ground truth SAE while leaving the encoder and decoder fixed at the correct features. We plot the variance explained vs L0 in Figure 4

for both SAEs. When the SAE L0 is lower than the true L0 of the toy model, the ground-truth SAE scores worse on reconstruction than the trained SAE! If we had an SAE training technique that gave us the ground truth correct SAE for a given LLM, sparsity–reconstruction plots would cause us to discard the correct SAE in favor of an incorrect SAE that mixes features together.

We show the cosine similarity of the SAE decoder latents with the ground truth features for the SAEs learned with L0=1 and L0=2 compared with the ground-truth SAE in Figure 5. Both these SAEs outperform the ground-truth SAE on variance explained by over 2x despite learning horribly polysemantic latents bearing little resemblance to the underlying true features of the model.

### 3.5 DETECTING THE TRUE L0 USING THE SAE DECODER

Figure 1 reveals that the SAE decoder latents contain mixes of underlying features, both when the L0 is too high and also when it is too low. As the SAE approaches the correct L0, each SAE latent has fewer components of multiple true features mixed in, becoming more monosemantic. Thus, we expect that the closer the SAE is to the correct L0, the more latents should be orthogonal relative to each other, as there are fewer components of shared correlated features mixed into latents. If we are far from the correct L0, then SAE latents contain components of many underlying features, and thus we expect latents to have higher cosine similarity with each other.

We call this metric *decoder pairwise cosine similarity*, $c_{\text{dec}}$, and define it as below:

$$c_{\text{dec}} = \frac{1}{\binom{h}{2}} \sum_{i=1}^{h-1} \sum_{j=i+1}^{h} |\cos(\mathbf{W}_{\text{dec},i}, \mathbf{W}_{\text{dec},j})| \tag{4}$$

where $\binom{h}{2} = \frac{h(h-1)}{2}$ is the total number of distinct pairs of latents in the SAE decoder.

If SAE decoder latents are mixing lots of positive and negative components of correlated and anti-correlated features, then each SAE latent should become less orthogonal to each other SAE latent, as many latents will likely mix together similar features. This should mean that the absolute value of the cosine similarity between arbitrary latents should also increase the worse this mixing becomes.

We calculate pairwise calculate similarity $c_{\text{dec}}$ for each of the BatchTopK SAEs we trained on toy models from Section 3.5. Results are shown in Figure 6. We see that pairwise cosine similarity is minimized at the true L0.

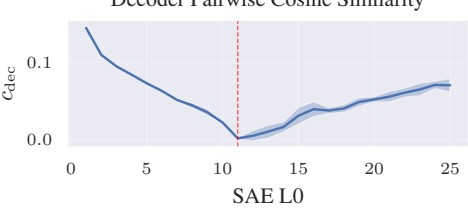

Figure 6: Decoder pairwise cosine similarity $c_{\text{dec}}$ evaluated on 5 seeds of toy model SAEs. The true L0 is indicated with a dotted line at 11. Shaded area is 1 stdev. $c_{\text{dec}}$ is minimized at the true L0.

We explore alternative metrics in Appendix A.9. Further toy model experiments are shown in Appendix A.4. Pytorch code implementing $c_{\text{dec}}$ is provided in Appendix A.17. We provide formal theoretical justification for the $c_{\text{dec}}$ metric in Appendix A.6.

### 3.6 JUMPRELU SAE EXPERIMENTS

So far, we have only investigated BatchTopK SAES due to their ease of setting L0. We now validate that these same conclusions apply to JumpReLU SAEs. We train JumpReLU saes with a range of $\lambda_s$ to control the sparsity of the SAEs. We show plots of $\lambda_s$ vs L0 and decoder pairwise cosine

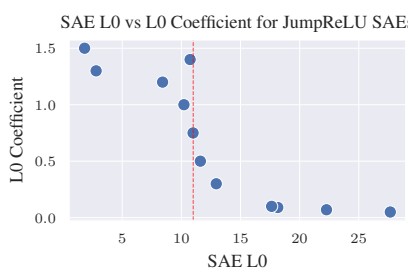 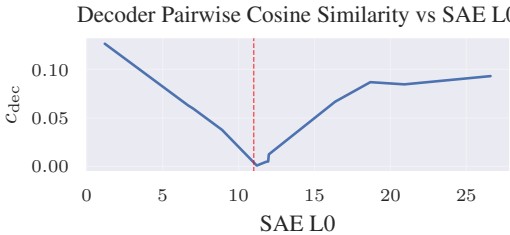

Figure 7: (left) L0 coefficient $\lambda_s$ vs L0 for JumpReLU SAEs. (right) Decoder pairwise cosine similarity vs L0 for JumpReLU SAEs. The true L0, 11, is marked by a dotted line on the plot.

Gemma-2-2B Layer 5 BatchTopK                    Llama-3.2-1B Layer 7 BatchTopK

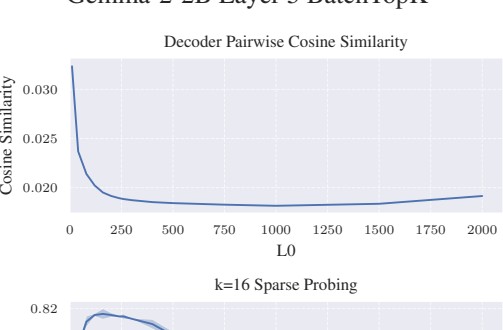 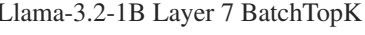 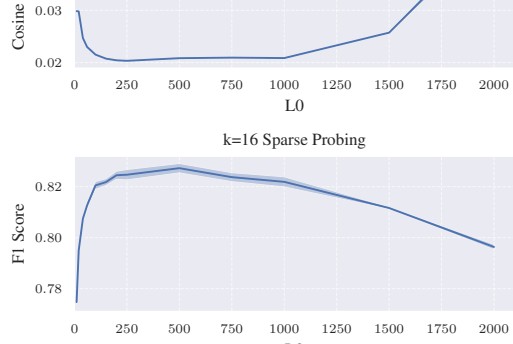

Figure 8: Decoder pairwise cosine similarity vs SAE L0 and K-sparse probing F1 vs L0 with 3 seeds per L0. (left) Gemma-2-2b layer 5 BatchTopK results. (right) Llama-3.2-1b layer 7 BatchTopK SAEs. In both cases, peak sparse probing performance occurs in the elbow just before $c_{\text{dec}}$ jumps due to low L0, although the shapes of the $c_{\text{dec}}$ plots vary at high L0.

similarity vs L0 for these SAEs in Figure 7. We see that the cosine similarity vs L0 broadly follows the same pattern as we saw for BatchTopK SAEs, and is minimized at the correct L0.

Interestingly, we see that the L0 does not change linearly with $\lambda_s$, but instead "sticks" near the correct L0. This is a testament to Anthropic's JumpReLU SAE training method (Conerly et al., 2025), as a wide range of sparsity coefficients $\lambda_s$ cause the SAE to naturally find the correct L0.

## 4   LLM EXPERIMENTS

We train a series of BatchTopK SAEs (Bussmann et al., 2024) with $h = 32768$ on Gemma-2-2b (Team et al., 2024) and Llama-3.2-1b (Dubey et al., 2024) varying L0 and calculate $c_{\text{dec}}$. Each SAE is trained on 500M tokens from the Pile (Gao et al., 2020) using SAELens (Bloom et al., 2024). We also calculate k-sparse probing performance for these SAEs using the benchmark from Kantamneni et al. (2025), consisting of over 100 sparse probing tasks. Results are shown in Figure 8.

The Llama SAE $c_{\text{dec}}$ plot looks very similar to the toy model, with a clear minimum point. The Gemma-2-2b layer 5 SAEs also show a sharp increase in $c_{\text{dec}}$ at low L0 as we saw in toy models, but has a long shallow region with the global minimum actually appearing in that shallow region. In both cases, the "elbow" in the $c_{\text{dec}}$ plots just before the jump due to low L0 is around L0 200, and

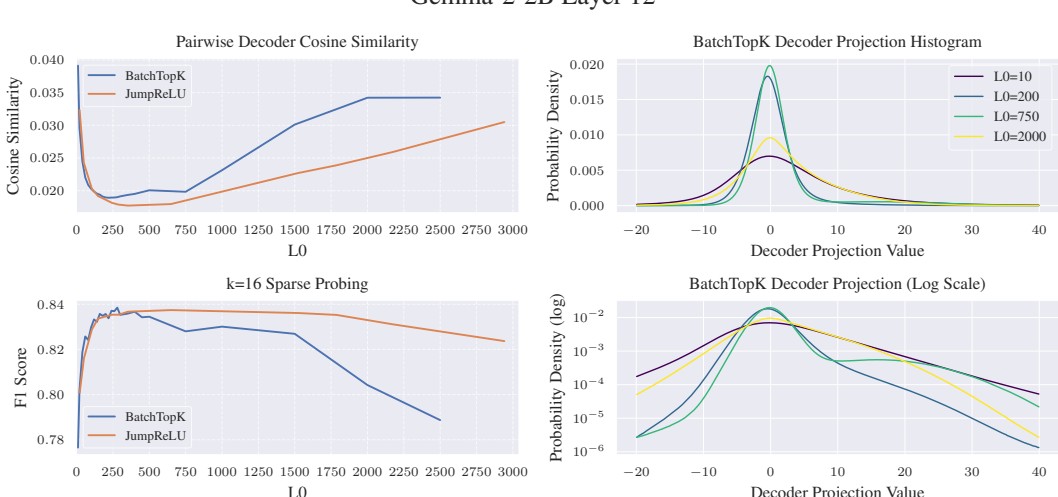

Figure 9: Gemma-2-2b layer 12, with (left) decoder pairwise cosine similarity and K-sparse probing F1 for BatchTopK and JumpReLU SAEs, and (right) normalized decoder projection histograms for BatchTopK SAEs. Histograms are truncated to -20 and 40 to highlight projections near the origin.

this also corresponds to peak sparse probing performance. More plots and analysis of $c_{\text{dec}}$ curves are shown in Appendix A.15.

## 4.1 JUMPRELU VS BATCHTOPK SAES

We next explore how JumpReLU and BatchTopK SAEs compare with decoder pairwise cosine similarity plots. We train a suite of SAEs on 1B tokens on Gemma-2-2b layer 12. We plot $c_{\text{dec}}$ for a range N values as well as k-sparse probing results for JumpReLU and BatchTopK SAEs in Figure 9 (left).

JumpReLU and BatchTopK SAEs behave similarly at low L0, with the high $c_{\text{dec}}$ at low L0 corresponding to poor sparse-probing performance. However, we see notable differences at high L0. The BatchTopK SAEs have a global $c_{\text{dec}}$ minimum around 200, but JumpReLU SAEs $c_{\text{dec}}$ minimum appears closer to 250-300. As we saw in Figure 8 as well, using the "elbow" of the plots just before $c_{\text{dec}}$ jumps due to low L0 seems to roughly correspond to peak k-sparse probing performance.

For JumpReLU SAEs, we see that $c_{\text{dec}}$ rises much less than BatchTopK SAEs at high L0, and indeed, JumpReLU SAEs also perform much better than BatchTopK SAEs at sparse probing when L0 is high. We suspect this is due to JumpReLU SAEs being able to "stick" near the correct threshold per latent like we saw in our toy models section. We investigate the differences in learned SAEs between JumpReLU and BatchTopK further in Appendix A.16.

## 4.2 CAN L0 BE BOTH TOO LOW AND TOO HIGH SIMULTANEOUSLY?

In Figure 9 (right), we plot decoder projection histogram plots for BatchTopK SAEs on Gemma-2-2b layer 12 with L0 10, 200, 750, and 2000. These plots are created by projecting training inputs on the SAE decoder, creating a histogram of how strongly each latents projects onto the input. We expect that the more SAE latents are mixing positive and negative components of underlying features, the more strongly they should project both positively and negatively on arbitrary training inputs. This should look like a narrow gaussian around 0 when there is little mixing, and a wider gaussian the more mixing there is. This is also the intuition behind the alternative metric discussed in Appendix A.9, and the theory behind this is formalized further in Appendix A.10.

As expected, when L0 is very low (10) or very high (2000), we see a wide gaussian around 0, indicating that decoder latents are mixing correlated features together. At L0=200, we see a much more narrow distribution around 0, as we expect when near the correct L0. However, at L0=750,

we see an interesting phenomenon, where there is an even narrower distribution than at L0=200, but also a large hump starting at projection above 10 (more visible in the log plot).

We suspect this indicates at L0=750, some latents become more monosemantic while other latents mix underlying features becoming less monosemantic. This likely means that the L0 is too high for some latents while simultaneously being too low for other latents. There is no reason why every latent has the same firing threshold, so there is likely a range of L0s where some latents are firing more than they ideally should while other latents are firing less than they ideally should. We also suspect this is part of why JumpReLU SAEs seem to perform much better at high L0, since JumpReLU SAEs can adjust firing threshold per-latent while BatchTopK SAEs cannot.

## 5 RELATED WORK

**Limitations of SAEs**   Early work on SAEs for interpretability highlight the problem of feature splitting (Bricken et al., 2023; Templeton et al., 2024), where a seemingly interpretable general feature splits into more specific features at narrower SAE widths. Chanin et al. (2025) explores feature hedging, showing SAEs mix correlated features into latents if the SAE is too narrow. We consider our work a version of feature hedging due to low L0. Till (2024) shows SAEs may increase sparsity by inventing features. Chanin et al. (2024) discuss the problem of feature absorption, where SAEs can improve their sparsity score by mixing hierarchical features together. Engels et al. (2024) investigates SAE errors and finds that SAE error may be pathological and non-linear. Engels et al. (2025) find that not all underlying LLM features themselves are linear, demonstrating circular embeddings of some concepts. Wu et al. (2025) and Kantamneni et al. (2025) both investigate empirical SAE performance, finding SAEs underperform relative to supervised baselines, but do not offer theoretical explanations as to why SAEs underperform.

**Picking SAE hyperparameters**   Related to our work is Minimum Description Lengths (MDL) SAEs (Ayonrinde et al., 2024), which attempt to find reasonable choices for SAE width and L0 based on information theory. However, MDL SAEs assume that there is no inherently "correct" decomposition for LLM activations and no "correct" L0, and therefore does not attempt to find the underlying true features. Our work takes the opposite approach, starting from simple toy models with linear features and showing that if L0 is not set correctly the SAE decoder becomes corrupted.

Another SAE architecture which attempts to pick L0 heuristically is Approximate Feature Activation (AFA) SAEs (Lee et al., 2025). AFA SAEs selects L0 adaptively at each input by assuming underlying true features are maximally orthogonal and selecting features until the feature norm is close to the input norm. While the L0 is not set directly in AFA SAEs, there is an extra loss hyperparameter that may modulate the resulting L0.

**Choosing L0 in related fields**   In Independent Component Analysis (ICA), a related field, it has also been shown that selecting the correct number of independent components (equivalent the L0 in SAEs) is important to achieve successful disentanglement (Li et al., 2007; Yi et al., 2024). However, ICA differs from SAEs in that the ICA requires fewer features than the number of input dimensions, while SAEs typically use overcomplete dictionaries.

## 6 DISCUSSION

While most practitioners of SAEs understand that having too high L0 is problematic, our work shows that having too low of L0 is perhaps even worse. Our work has several important implications for the field. First, the L0 used by most SAEs is lower than it ideally should be, as a cursory search of open source SAEs on Neuronpedia (Lin, 2023) shows L0 less than 100 is very common even for SAEs trained on large models (see Appendix A.13). We further show that the sparsity–reconstruction tradeoff, as commonly discussed by most SAE papers (Cunningham et al., 2024; Gao et al., 2024; Rajamanoharan et al., 2024), is misleading: when L0 is too low, an SAE with a correct dictionary achieves worse reconstruction than an incorrect SAE that mixes correlated features.

We presented a metric based on the correlation between the SAE decoder and input activations, $c_{\text{dec}}$, that can give us hints about the correct L0 for a given SAE. However, we do not view this as a

perfect guide. As we saw in our results, while low L0 SAEs consistently have very high $c_{dec}$, the metric can sometime remain nearly flat for a wide range of L0. Still, we feel that this metric is a useful guide to avoid L0 that is clearly too low, and we hope this investigation into correlation-based SAE quality metrics can be built on further in future work. We are particularly excited about the possibility that we can learn more about the underlying correlational structure between underlying features by studying correlations in the SAE decoder.

While our metric currently requires training a sweep over L0 to optimize, we are hopeful that it may be possible to optimize this metric automatically during training (steps towards this are discussed in Appendix A.11). Improving this further is left to future work.

## 7 REPRODUCIBILITY STATEMENT

Code for all toy model experiments and demonstration code for training and evaluating LLM SAEs is provided as part of the supplementary materials for this paper. We further provide details on toy model SAE training in Section 3 and Appendix A.2, and for LLM SAE training Section 4 and Appendix A.7.

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

# A    APPENDIX

## A.1    SAE TRAINING ARCHITECTURE DEFINITIONS

In this work we focus on JumpReLU (Conerly et al., 2025; Rajamanoharan et al., 2024) and Batch-TopK (Bussmann et al., 2024) SAEs. For BatchTopK SAEs, there is no sparsity penalty as sparsity is enforced by the BatchTopK function. The auxiliary loss $\mathcal{L}_P$ for BatchTopK is as follows, where $e$ is the SAE training error residual, and $\hat{e}$ is a reconstruction using the top $k_{\text{aux}}$ dead latents (meaning the latents have not fired in more than $n_{\text{dead}}$ steps during training).

$$\mathcal{L}_p = \|e - \hat{e}\|_2^2$$

We follow the JumpReLU training setup from Conerly et al. (2025), which involves both a sparsity loss $\mathcal{L}_s$ and a pre-activation loss for reviving dead latents, $\mathcal{L}_p$. $\mathcal{L}_s$ is defined as below, where $c$ is a scalar scaling factor:

$$\mathcal{L}_s = \sum_i \tanh(c * |a_i| \|\mathbf{W}_{\text{dec},i}\|_2)$$

The pre-activation loss $\mathcal{L}_p$ adds a small penalty for all dead features, where $a_{\text{pre}}$ refers to the pre-activation of the SAE passed into the JumpReLU:

$$\mathcal{L}_p = \sum_i \text{ReLU}(\tau_i - a_{\text{pre},i}) \|\mathbf{W}_{\text{dec},i}\|_2$$

The JumpReLU defines a pseudo-gradient relative to the threshold $\tau$ as follows, where $\epsilon$ is the bandwidth of the estimator:

$$\frac{\partial \text{JumpReLU}(x,\tau)}{\partial \tau} = \begin{cases} -\frac{\tau}{\epsilon} & \text{if } -\frac{1}{2} < \frac{x-\tau}{\epsilon} < \frac{1}{2} \\ 0 & \text{otherwise} \end{cases}$$

## A.2    TOY MODEL SAE TRAINING DETAILS

We train on 15M samples with a batch size of 1024 for all toy model experiments, and a learning rate of 3e-4. We do not use any learning rate warm-up or decay. For all SAE latents vs true feature cosine similarity plots, we re-arrange the SAE latents so the latent indices match the feature indices in the plots, as this makes interpreting the plots easier without any loss of generality.

For the large toy model experiments in Section 3.2, we use a randomly generated correlation matrix and linearly decreasing feature firing probabilities, both shown in Figure 10.

## A.3    EXTENDED SMALL TOY MODEL EXPERIMENTS

We continue our investigation of feature mixing due to low L0 and correlated features using the same five-feature toy model from Section 3.1.

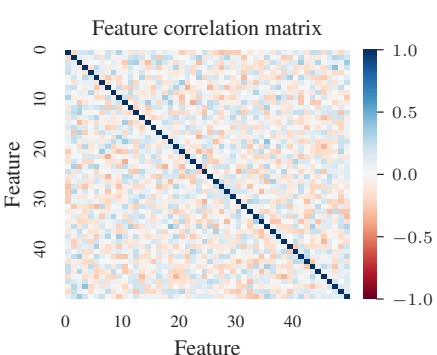 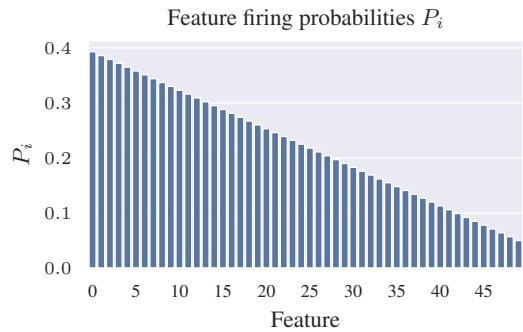

Figure 10: (left) random correlation matrix and (right) base feature firing probabilities for toy model.

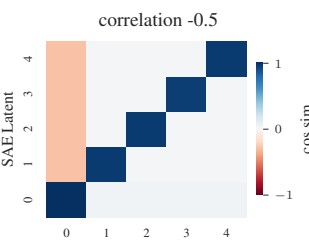 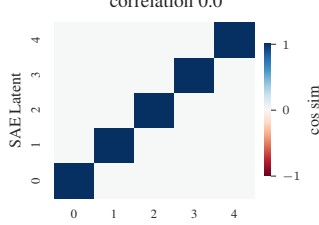 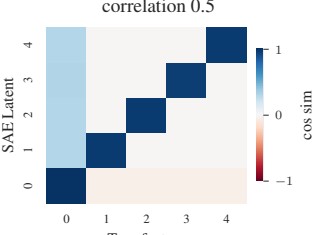

Figure 11: Decoder cosine similarity with true features for SAEs trained on toy models with different amounts of correlation between $f_0$ and features 1-4. When there is negative correlation (left), we see the SAE mix in a negative component of $f_0$ into latents 1-4. When there is positive correlation (right), we see a positive component of $f_0$ mixed into the SAE latents. When there is no correlation (middle), there is no mixing.

### A.3.1 VARYING FEATURE CORRELATION STRENGTH

We now explore the effect of varying the strength of the correlation between feature $f_0$ and features $f_1$, $f_2$, $f_3$ and $f_4$. In our earlier experiments, we used correlation of $0.4$ and $-0.4$. Here, we will vary correlation between $-0.5$ and $0.5$, while keeping L0=1.8, lower than the true L0 of 2.0. We show decoder cosine similarity plot with true features with correlation $-0.5$, $0.0$, and $0.5$ in Figure 11. As expected, latents 1-4 mix negative components of $f_0$ when correlation is negative, positive components of $f_0$ when correlation is positive, and we see no mixing at all when there is no correlation.

We next measure the amount of mixing by calculating the mean cosine similarity of feature 0 with the SAE latents tracking features $f_1$ through $f_4$. We show results in Figure 12. As expected the more negative the correlation, the more negative the mixing. The more positive the correlation, the more positive the mixing. When there is no correlation, there is no mixing.

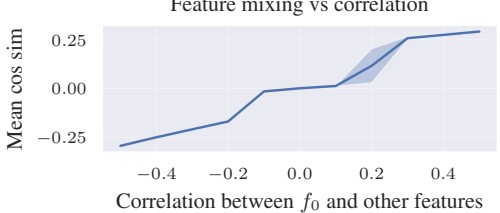

Figure 12: Amount of mixing (measured as mean cosine similarity between SAE latents 1-4 and feature 0) vs correlation between feature 0 and features 1-4.

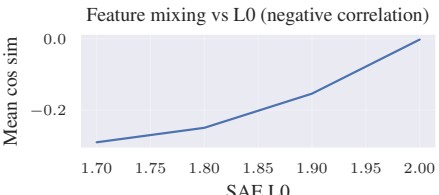 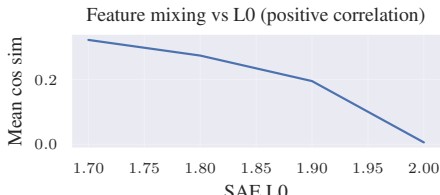

Figure 13: Amount of mixing (measured as mean cosine similarity between SAE latents 1-4 and feature 0) vs SAE L0 (true L0 is 2.0). (Left) mixing vs L0 with negative correlation, and (right) mixing vs L0 with positive correlation.

### A.3.2 VARYING L0

Next, we study how varying the L0 of the SAE affects the amount of feature mixing we observe. We fix the correlation between feature 0 and features 1-4 at 0.4 for positive correlation, and -0.4 for negative correlation, as in Section 3.1, and vary the L0 of the SAE from 1.7 to 2.0 (the true L0 of the toy model is 2.0). We find that dropping the L0 below 1.7 causes the SAE latents to become so deformed that they bear almost no resemblance to the true features, making it difficult to perform systematic analysis.

Results are shown in Figure 13. As expected, the further the SAE L0 gets from the true L0 (2.0), the worse the mixing becomes. Furthermore, the mixing matches the sign of the correlation, with negative correlation causing negative mixing, and positive correlation causing positive mixing.

### A.3.3 SUPERPOSITION NOISE

We have showed that even in the simplest possible setting for an SAE with perfect orthogonality between features, SAEs will fail to learn true features if the SAE L0 is too low and there is correlation between features. If there is superposition noise making the task harder for the SAE, there is thus no reason to expect that SAE will somehow perform better. Regardless, we include results on a toy model with superposition noise below for completeness, as in we use SAEs in situations with superposition noise.

For this experiment, we reuse the same positive and negative correlations from Section 3.1 (+0.4 and -0.4). However, we allow the toy model features to have small positive and negative overlap with each other. We then train an SAE on this toy model with L0=1.9. We find that the using the previous L0=1.8 breaks the SAE too much given the added challenge of superposition noise. We show results in Figure 14.

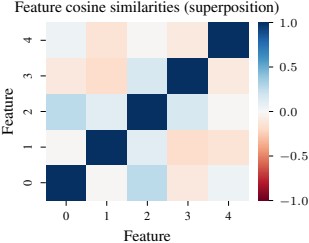 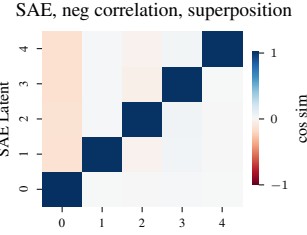 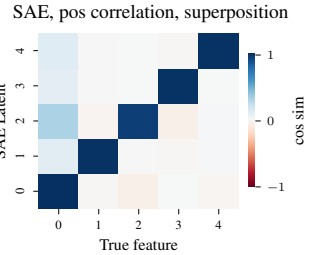

Figure 14: Superposition toy model results. For the SAE decoder cosine similarity plots, we subtract out the cosine similarity of the underlying features due to superposition for clarity. (Left) cosine similarity between underlying toy model features, showing positive and negative overlaps between features. (Middle) SAE decoder similarity with true feature with negative correlation between feature 0 and features 1-4. (Right) SAE decoder cosine similarity with true features with positive correlation between feature 0 and features 1-4.

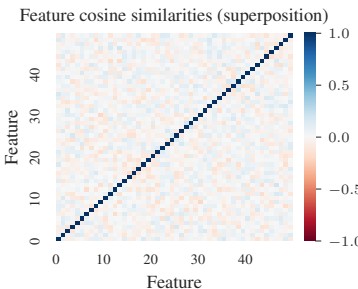
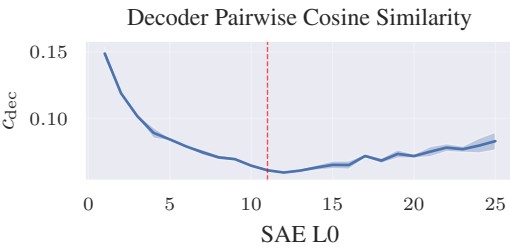

Figure 15: (Left) cosine similarity between the true features in the toy model. Due to superposition noise, each features overlaps slightly with many other features. (Right) decoder pairwise cosine similarity for SAEs trained at different L0s. The true L0 is marked with a dashed line.

We see the same pattern as we saw with no superposition noise: the SAE mixes correlated features based on the sign of the correlation. The superposition noise has made the results a bit noisier, but the core trend is still clearly visible.

### A.4 EXTENDED LARGE TOY MODEL EXPERIMENTS

In this section, we build on the results from the 50-latent toy model from Section 3.2.

#### A.4.1 SUPERPOSITION NOISE

We now modify the large toy model to have superposition noise, as this is a more realistic setting for an LLM SAE to operate in. We reducing the dimensionality of the space to 40, lower than the number of features in the toy model (50). This forces each feature to slightly overlap other features in the space. The resulting feature cosine similarities are shown in Figure 15 (left).

We train 5 seeds of SAEs at a range of L0s on this superposition toy model, and calculate $c_{\text{dec}}$ in Figure 15 (right). We see that decoder pairwise cosine similarity is still roughly minimized at the true L0 of the toy model.

### A.5 PROOF: LOW L0 INCENTIVIZES FEATURE MIXING

We now provide a theoretical proof that when SAE L0 is less than the true L0 of the underlying features, MSE loss directly incentivizes the SAE to mix features together.

**Theorem 1.** *Consider a toy model with two orthonormal features $\mathbf{f}_1, \mathbf{f}_2 \in \mathbb{R}^d$ where $\mathbf{f}_1 \cdot \mathbf{f}_2 = 0$ and $\|\mathbf{f}_1\|_2 = \|\mathbf{f}_2\|_2 = 1$. Let $\mathbf{f}_1$ fire alone with probability $P_1$, $\mathbf{f}_2$ fire alone with probability $P_2$, and both fire together with probability $P_{12}$, where $P_1 + P_2 + P_{12} \leq 1$. Consider a tied SAE with 2 latents (i.e., $\mathbf{W}_{enc} = \mathbf{W}_{dec}^T$) and no biases that can fire at most 1 latent per input (L0 = 1). We assume this is less than the L0 of the data (i.e., $\mathbb{E}[\text{active features}] = P_1 + P_2 + 2P_{12} > 1$), which occurs whenever features co-occur ($P_{12} > 0$). Then the SAE that minimizes expected MSE will have latents that mix $\mathbf{f}_1$ and $\mathbf{f}_2$, rather than learning them separately.*

*Proof.* We define our SAE with decoder $\mathbf{W}_{\text{dec}} = [\mathbf{l}_1, \mathbf{l}_2] \in \mathbb{R}^{d \times 2}$ where $\mathbf{l}_1, \mathbf{l}_2$ are the two latent directions. Since the SAE is tied and has no biases, the reconstruction of an input $\mathbf{x}$ using a single active latent $\mathbf{l}_i$ (selected via Top-1 projection) is:

$$\hat{\mathbf{x}} = (\mathbf{l}_i \cdot \mathbf{x})\mathbf{l}_i \tag{5}$$

The reconstruction loss for a single sample is:

$$\mathcal{L}(\mathbf{x}) = \|\mathbf{x} - \hat{\mathbf{x}}\|_2^2 = \|\mathbf{x} - (\mathbf{l}_i \cdot \mathbf{x})\mathbf{l}_i\|_2^2 \tag{6}$$

**Parameterization.** We parameterize the latents as:

$$\mathbf{l}_2 = \mathbf{f}_2 \tag{7}$$

$$\mathbf{l}_1 = \frac{\alpha\mathbf{f}_1 + (1-\alpha)\mathbf{f}_2}{\sqrt{\alpha^2 + (1-\alpha)^2}} \tag{8}$$

where $0 \le \alpha \le 1$ controls the mixture. When $\alpha = 1$, $\mathbf{l}_1 = \mathbf{f}_1$ (the correct, disentangled solution). When $0 \le \alpha < 1$, $\mathbf{l}_1$ mixes both features.

**Case analysis.** We analyze the four possible cases:

**Case 1:** Only $\mathbf{f}_1$ fires (probability $P_1$). The input is $\mathbf{x} = m_1\mathbf{f}_1$ where $m_1 > 0$ is the magnitude. Latent $\mathbf{l}_1$ activates (since it has the largest projection). The reconstruction loss is:

$$\mathcal{L}_1(\alpha) = \|m_1\mathbf{f}_1 - (\mathbf{l}_1 \cdot m_1\mathbf{f}_1)\mathbf{l}_1\|_2^2 \tag{9}$$

$$= \left\| m_1\mathbf{f}_1 - \frac{m_1\alpha}{\sqrt{\alpha^2 + (1-\alpha)^2}} \cdot \frac{\alpha\mathbf{f}_1 + (1-\alpha)\mathbf{f}_2}{\sqrt{\alpha^2 + (1-\alpha)^2}} \right\|_2^2 \tag{10}$$

$$= \left\| m_1\mathbf{f}_1 - \frac{m_1\alpha^2}{\alpha^2 + (1-\alpha)^2}\mathbf{f}_1 - \frac{m_1\alpha(1-\alpha)}{\alpha^2 + (1-\alpha)^2}\mathbf{f}_2 \right\|_2^2 \tag{11}$$

$$= m_1^2 \left[ \left( 1 - \frac{\alpha^2}{\alpha^2 + (1-\alpha)^2} \right)^2 + \left( \frac{\alpha(1-\alpha)}{\alpha^2 + (1-\alpha)^2} \right)^2 \right] \tag{12}$$

Simplifying using $\alpha^2 + (1-\alpha)^2 = 1 - 2\alpha(1-\alpha)$:

$$\mathcal{L}_1(\alpha) = m_1^2 \left[ \left( \frac{(1-\alpha)^2}{\alpha^2 + (1-\alpha)^2} \right)^2 + \left( \frac{\alpha(1-\alpha)}{\alpha^2 + (1-\alpha)^2} \right)^2 \right] \tag{13}$$

$$= m_1^2 \cdot \frac{(1-\alpha)^2[\alpha^2 + (1-\alpha)^2]}{[\alpha^2 + (1-\alpha)^2]^2} \tag{14}$$

$$= m_1^2 \cdot \frac{(1-\alpha)^2}{\alpha^2 + (1-\alpha)^2} \tag{15}$$

**Case 2:** Only $\mathbf{f}_2$ fires (probability $P_2$). The input is $\mathbf{x} = m_2\mathbf{f}_2$. Latent $\mathbf{l}_2 = \mathbf{f}_2$ activates, giving perfect reconstruction:

$$\mathcal{L}_2 = 0 \tag{16}$$

**Case 3:** Both $\mathbf{f}_1$ and $\mathbf{f}_2$ fire (probability $P_{12}$). The input is $\mathbf{x} = m_1\mathbf{f}_1 + m_2\mathbf{f}_2$. Since L0 = 1, only one latent can activate. The SAE will choose $\mathbf{l}_1$ if $|\mathbf{l}_1 \cdot \mathbf{x}|^2 > |\mathbf{l}_2 \cdot \mathbf{x}|^2$. We have:

$$|\mathbf{l}_1 \cdot \mathbf{x}|^2 = \left( \frac{m_1\alpha + m_2(1-\alpha)}{\sqrt{\alpha^2 + (1-\alpha)^2}} \right)^2 = \frac{(m_1\alpha + m_2(1-\alpha))^2}{\alpha^2 + (1-\alpha)^2} \tag{17}$$

$$|\mathbf{l}_2 \cdot \mathbf{x}|^2 = m_2^2 \tag{18}$$

For simplicity, we assume $m_1 \ge m_2 > 0$, so $\mathbf{l}_1$ will activate when $\alpha$ is sufficiently large (e.g., for $\alpha = 1$, $|\mathbf{l}_1 \cdot \mathbf{x}|^2 = m_1^2 > m_2^2$). Assuming $\mathbf{l}_1$ activates, the reconstruction loss is:

$$\mathcal{L}_3(\alpha) = \|m_1\mathbf{f}_1 + m_2\mathbf{f}_2 - (\mathbf{l}_1 \cdot (m_1\mathbf{f}_1 + m_2\mathbf{f}_2))\mathbf{l}_1\|_2^2 \tag{19}$$

$$= \left\| m_1\mathbf{f}_1 + m_2\mathbf{f}_2 - \frac{m_1\alpha + m_2(1-\alpha)}{\sqrt{\alpha^2 + (1-\alpha)^2}} \cdot \frac{\alpha\mathbf{f}_1 + (1-\alpha)\mathbf{f}_2}{\sqrt{\alpha^2 + (1-\alpha)^2}} \right\|_2^2 \tag{20}$$

Let $c = \frac{m_1\alpha + m_2(1-\alpha)}{\alpha^2 + (1-\alpha)^2}$. Then:

$$\mathcal{L}_3(\alpha) = \|m_1\mathbf{f}_1 + m_2\mathbf{f}_2 - c\alpha\mathbf{f}_1 - c(1-\alpha)\mathbf{f}_2\|_2^2 \tag{21}$$

$$= (m_1 - c\alpha)^2 + (m_2 - c(1-\alpha))^2 \tag{22}$$

Expanding and simplifying (see detailed algebra below):

$$\mathcal{L}_3(\alpha) = \frac{[m_1(1-\alpha) - m_2\alpha]^2}{\alpha^2 + (1-\alpha)^2} \tag{23}$$

Note that when $m_1 = m_2 = m$, this simplifies to:

$$\mathcal{L}_3(\alpha) = \frac{m^2(1-2\alpha)^2}{\alpha^2 + (1-\alpha)^2} \tag{24}$$

which equals 0 when $\alpha = 0.5$ (perfect reconstruction when features are equally mixed) and equals $m^2$ when $\alpha = 1$ (complete failure to reconstruct $\mathbf{f}_2$).

**Case 4:** Neither feature fires (probability $P_0 = 1 - P_1 - P_2 - P_{12}$). Perfect reconstruction with $\mathcal{L}_4 = 0$.

**Expected loss.** The expected loss is:

$$\mathbb{E}[\mathcal{L}(\alpha)] = P_1 \mathbb{E}_{m_1}[\mathcal{L}_1(\alpha)] + P_2 \mathbb{E}_{m_2}[\mathcal{L}_2(\alpha)] + P_{12} \mathbb{E}_{m_1, m_2}[\mathcal{L}_3(\alpha)] \tag{25}$$

Assuming $\mathbf{l}_1$ activates in Case 1 and $\mathbf{l}_2$ in Case 2 (which holds for $\alpha > 0.5$):

$$\mathbb{E}[\mathcal{L}(\alpha)] = P_1 \mathbb{E}_{m_1}\left[m_1^2 \frac{(1-\alpha)^2}{\alpha^2 + (1-\alpha)^2}\right] + P_{12} \mathbb{E}_{m_1, m_2}\left[\frac{[m_1(1-\alpha) - m_2\alpha]^2}{\alpha^2 + (1-\alpha)^2}\right] \tag{26}$$

**Concrete example demonstrating feature mixing.** To make this concrete, suppose $m_1 = m_2 = 1$ (both features have equal magnitude when they fire). Assume equal probabilities $P_1 = P_{12} = 0.4$, which implies $P_2 = 0$ (or is negligible) and $P_0 = 0.2$.

For the disentangled solution ($\alpha = 1$, so $\mathbf{l}_1 = \mathbf{f}_1$):

$$\mathcal{L}_1(\alpha = 1) = 0 \quad \text{(perfect reconstruction when only } \mathbf{f}_1 \text{ fires)} \tag{27}$$

$$\mathcal{L}_3(\alpha = 1) = \frac{m^2(1-2)^2}{1^2 + 0^2} = m^2 = 1 \quad \text{(cannot reconstruct } \mathbf{f}_2 \text{ component)} \tag{28}$$

Expected loss: $\mathbb{E}[\mathcal{L}(\alpha = 1)] = (0.4 \times 0) + (0.4 \times 1) = 0.4$

For a mixed solution ($\alpha = 0.6$):

$$\mathcal{L}_1(\alpha = 0.6) = 1^2 \cdot \frac{(1-0.6)^2}{0.6^2 + 0.4^2} = \frac{0.16}{0.52} \approx 0.308 \tag{29}$$

$$\mathcal{L}_3(\alpha = 0.6) = 1^2 \cdot \frac{(1 - 2 \times 0.6)^2}{0.6^2 + 0.4^2} = \frac{(-0.2)^2}{0.52} = \frac{0.04}{0.52} \approx 0.077 \tag{30}$$

Expected loss: $\mathbb{E}[\mathcal{L}(\alpha = 0.6)] = (0.4 \times 0.308) + (0.4 \times 0.077) \approx 0.1232 + 0.0308 = 0.154$

The mixed solution achieves $\mathbb{E}[\mathcal{L}(\alpha = 0.6)] \approx 0.154 < 0.4 = \mathbb{E}[\mathcal{L}(\alpha = 1)]$, demonstrating that MSE loss directly incentivizes feature mixing when L0 is constrained below the true L0.

**Optimal mixing coefficient.** More generally, for the case $m_1 = m_2 = m$, the expected loss is:

$$\mathbb{E}[\mathcal{L}(\alpha)] = \frac{m^2}{\alpha^2 + (1-\alpha)^2}\left[P_1(1-\alpha)^2 + P_{12}(1-2\alpha)^2\right] \tag{31}$$

At the boundaries:

- At $\alpha = 1$ (disentangled): $\mathbb{E}[\mathcal{L}(1)] = P_{12}m^2$

- At $\alpha = 0.5$ (maximally mixed): $\mathbb{E}[\mathcal{L}(0.5)] = \frac{P_1 m^2 (0.5)^2}{0.5^2 + 0.5^2} = \frac{P_1 m^2 (0.25)}{0.5} = \frac{P_1 m^2}{2}$

When $P_{12} > P_1/2$, we have $\mathbb{E}[\mathcal{L}(0.5)] < \mathbb{E}[\mathcal{L}(1)]$, showing that mixing reduces loss when both features frequently co-occur. For instance, with $P_1 = P_{12} = 0.5$ and $m = 1$:

$$\mathbb{E}[\mathcal{L}(\alpha = 1)] = 0.5 \tag{32}$$
$$\mathbb{E}[\mathcal{L}(\alpha = 0.5)] = 0.25 \tag{33}$$

This demonstrates that when features frequently co-occur ($P_{12}$ is large), the MSE-optimal solution involves substantial feature mixing ($\alpha^* < 1$) rather than learning them disentangled ($\alpha = 1$), completing the proof. □

**Detailed algebra for Case 3.** Starting from:

$$\mathcal{L}_3(\alpha) = (m_1 - c\alpha)^2 + (m_2 - c(1 - \alpha))^2 \tag{34}$$

where $c = \frac{m_1\alpha + m_2(1 - \alpha)}{\alpha^2 + (1 - \alpha)^2}$.

Expanding:

$$\mathcal{L}_3 = m_1^2 - 2m_1c\alpha + c^2\alpha^2 + m_2^2 - 2m_2c(1 - \alpha) + c^2(1 - \alpha)^2 \tag{35}$$
$$= m_1^2 + m_2^2 + c^2[\alpha^2 + (1 - \alpha)^2] - 2c[m_1\alpha + m_2(1 - \alpha)] \tag{36}$$

Note that $c[\alpha^2 + (1 - \alpha)^2] = m_1\alpha + m_2(1 - \alpha)$ by definition of $c$. Therefore:

$$\mathcal{L}_3 = m_1^2 + m_2^2 + c[m_1\alpha + m_2(1 - \alpha)] - 2c[m_1\alpha + m_2(1 - \alpha)] \tag{37}$$
$$= m_1^2 + m_2^2 - c[m_1\alpha + m_2(1 - \alpha)] \tag{38}$$
$$= m_1^2 + m_2^2 - \frac{[m_1\alpha + m_2(1 - \alpha)]^2}{\alpha^2 + (1 - \alpha)^2} \tag{39}$$

Further simplification:

$$\mathcal{L}_3 = \frac{(m_1^2 + m_2^2)[\alpha^2 + (1 - \alpha)^2] - [m_1\alpha + m_2(1 - \alpha)]^2}{\alpha^2 + (1 - \alpha)^2} \tag{40}$$

The numerator expands to:

$$(m_1^2 + m_2^2)[\alpha^2 + (1 - \alpha)^2] - [m_1^2\alpha^2 + 2m_1m_2\alpha(1 - \alpha) + m_2^2(1 - \alpha)^2] \tag{41}$$
$$= m_1^2\alpha^2 + m_1^2(1 - \alpha)^2 + m_2^2\alpha^2 + m_2^2(1 - \alpha)^2 - m_1^2\alpha^2 - 2m_1m_2\alpha(1 - \alpha) - m_2^2(1 - \alpha)^2 \tag{42}$$
$$= m_1^2(1 - \alpha)^2 + m_2^2\alpha^2 - 2m_1m_2\alpha(1 - \alpha) \tag{43}$$
$$= [m_1(1 - \alpha) - m_2\alpha]^2 \tag{44}$$

We can verify this factorization:

$$[m_1(1 - \alpha) - m_2\alpha]^2 = m_1^2(1 - \alpha)^2 - 2m_1m_2\alpha(1 - \alpha) + m_2^2\alpha^2 \tag{45}$$

This matches. Therefore:

$$\mathcal{L}_3(\alpha) = \frac{[m_1(1 - \alpha) - m_2\alpha]^2}{\alpha^2 + (1 - \alpha)^2} \tag{46}$$

A.6 THEORETICAL JUSTIFICATION FOR $c_{\text{DEC}}$ METRIC

We provide a theoretical justification for why the decoder pairwise cosine similarity metric, $c_{\text{dec}}$, serves as a proxy for detecting feature mixing in SAEs.

**Theorem 2.** *Consider two SAEs with identical dictionary size $h$, where SAE 1 learns disentangled features and SAE 2 mixes a correlated feature into its latents. Let the underlying true features $\{\mathbf{f}_1, \ldots, \mathbf{f}_h, \mathbf{g}\}$ be an orthonormal set in $\mathbb{R}^d$, where $\mathbf{f}_i$ are unique features and $\mathbf{g}$ is a dense or*

*frequent feature correlated with multiple $\mathbf{f}_i$. We model the decoder weights $\mathbf{W}_{dec,i}$ (normalized to unit length) for the two SAEs as:*

$$\text{SAE 1 (Disentangled):} \quad \mathbf{W}_i^{(1)} = \mathbf{f}_i \tag{47}$$

$$\text{SAE 2 (Mixed):} \quad \mathbf{W}_i^{(2)} = \sqrt{1-\gamma_i^2}\mathbf{f}_i + \gamma_i\mathbf{g} \tag{48}$$

*where $\gamma_i \in [-1,1]$ represents the mixing coefficient for latent $i$. Assume there exists a subset of latents $S \subseteq \{1,\ldots,h\}$ with $|S| \geq 2$ such that for all $i \in S$, $\gamma_i \neq 0$. Then, the expected pairwise cosine similarity is strictly greater for SAE 2 than SAE 1:*

$$c_{dec}(\text{SAE 2}) > c_{dec}(\text{SAE 1}) \tag{49}$$

*Proof.* Recall the definition of decoder pairwise cosine similarity:

$$c_{\text{dec}} = \frac{1}{\binom{h}{2}}\sum_{i=1}^{h-1}\sum_{j=i+1}^{h}|\cos(\mathbf{W}_{\text{dec},i}, \mathbf{W}_{\text{dec},j})| \tag{50}$$

Since the decoder weights are normalized, $\cos(\mathbf{W}_{\text{dec},i}, \mathbf{W}_{\text{dec},j}) = \mathbf{W}_{\text{dec},i}^{\top}\mathbf{W}_{\text{dec},j}$.

**Case 1: SAE 1 (Disentangled).** For any distinct pair $i \neq j$, the weights are $\mathbf{W}_i^{(1)} = \mathbf{f}_i$ and $\mathbf{W}_j^{(1)} = \mathbf{f}_j$. Since the underlying features are orthonormal:

$$\mathbf{W}_i^{(1)\top}\mathbf{W}_j^{(1)} = \mathbf{f}_i^{\top}\mathbf{f}_j = 0 \tag{51}$$

Thus, for SAE 1:

$$c_{\text{dec}}(\text{SAE 1}) = 0 \tag{52}$$

**Case 2: SAE 2 (Mixed).** Consider the dot product for a distinct pair $i, j$:

$$\mathbf{W}_i^{(2)\top}\mathbf{W}_j^{(2)} = (\sqrt{1-\gamma_i^2}\mathbf{f}_i + \gamma_i\mathbf{g})^{\top}(\sqrt{1-\gamma_j^2}\mathbf{f}_j + \gamma_j\mathbf{g}) \tag{53}$$

$$= \sqrt{(1-\gamma_i^2)(1-\gamma_j^2)}(\mathbf{f}_i^{\top}\mathbf{f}_j) + \gamma_j\sqrt{1-\gamma_i^2}(\mathbf{f}_i^{\top}\mathbf{g})$$

$$+ \gamma_i\sqrt{1-\gamma_j^2}(\mathbf{g}^{\top}\mathbf{f}_j) + \gamma_i\gamma_j(\mathbf{g}^{\top}\mathbf{g}) \tag{54}$$

Using the orthonormality of the set $\{\mathbf{f}_1,\ldots,\mathbf{f}_h,\mathbf{g}\}$:

- $\mathbf{f}_i^{\top}\mathbf{f}_j = 0$

- $\mathbf{f}_i^{\top}\mathbf{g} = 0$ and $\mathbf{g}^{\top}\mathbf{f}_j = 0$

- $\mathbf{g}^{\top}\mathbf{g} = 1$

The expression simplifies to:

$$\cos(\mathbf{W}_i^{(2)}, \mathbf{W}_j^{(2)}) = \gamma_i\gamma_j \tag{55}$$

The metric $c_{\text{dec}}$ is the average of absolute cosine similarities:

$$c_{\text{dec}}(\text{SAE 2}) = \frac{1}{\binom{h}{2}}\sum_{i<j}|\gamma_i\gamma_j| \tag{56}$$

Since we assumed there exists a subset $S$ where $\gamma_i \neq 0$, there exists at least one pair $(i,j)$ where $|\gamma_i\gamma_j| > 0$. All other terms are non-negative. Therefore:

$$c_{\text{dec}}(\text{SAE 2}) > 0 = c_{\text{dec}}(\text{SAE 1}) \tag{57}$$

This confirms that mixing a shared feature component into multiple latents strictly increases the $c_{\text{dec}}$ metric. $\square$

**Remark 1.** *In real-world scenarios with superposition noise, the baseline orthogonality $\mathbf{f}_i^{\top}\mathbf{f}_j$ is not exactly zero but follows a distribution with mean zero and variance $\approx 1/d$. However, systematic feature mixing introduces a structured non-zero component ($\gamma_i\gamma_j$) that typically dominates the random superposition noise, causing a measurable rise in $c_{dec}$ as observed in Figure 6 and Figure 8.*

### A.7 LLM SAE TRAINING DETAILS

For BatchTopK SAEs, we ensure that the decoder remains normalized with $||\mathbf{W}_{\text{dec}}||_2 = 1$ so $s_n^{\text{dec}}$ calculations use the same scale for every latent. We use a learning rate of $3e^{-4}$ with no warmup or decay.

For JumpReLU SAEs, we broadly follow the training procedure laid out by Conerly et al. (2025). However, we do not apply learning rate decay, and only warm $\lambda_s$ for 100M tokens to avoid the sparsity penalty changing throughout the majority of training. We use a learning rate of $2e^{-4}$, $c = 4$, $\lambda_p = 3e^{-6}$ and bandwidth $\epsilon = 2.0$ as recommended by (Conerly et al., 2025).

### A.8 TRANSITIONING L0 DURING TRAINING

We explore the effect of transitioning the L0 of the SAE during training using the toy model from Section 3. This toy model has a true L0 of 11. We train BatchTopK SAEs with a final L0 of 11, but starting with L0 either too high or too low, and linearly transitioning to the correct L0 over the first 25k steps of training, leaving the SAE at the correct L0 for the final 5k steps of training. We use a starting L0 of 20 for the case where we start too high, and use a starting L0 of 2 for the case where we start too low. Results are shown in Figure 16.

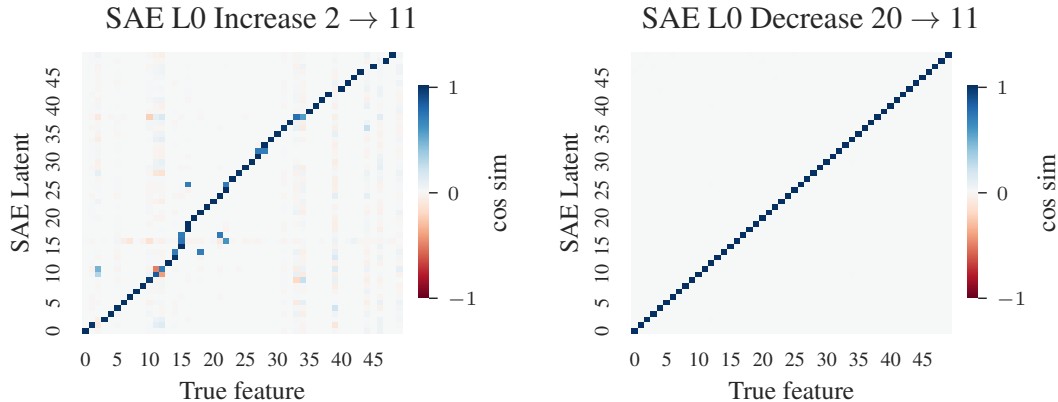

Figure 16: Transitioning L0 from too low (left) and too high (right) to the correct L0 during training. When the starting L0 is too high, the SAE still learns the correct features at the end of training. However, when L0 is too low, the SAE cannot recover fully and still learns many incorrect features at the end of training.

We see that decreasing the L0 of the SAE from a too high value to the correct value still results in the SAE learning correct features. However, when the SAE starts from a too low L0, the SAE cannot fully recover when the L0 is adjusted to the correct value later. It seems that the latents the SAE learns when L0 is too low is a local minimum that is difficult from the SAE to escape from even when the L0 is later corrected. This is likely because the latents learned when L0 is too low are optimized by gradient descent to achieve a higher MSE loss than is achievable by the correct latents under the same L0 constraint. However, when L0 is too high, there is no equivalent optimization pressure, and is thus less likely to be a local minimum.

### A.9 ALTERNATIVE METRIC: N^TH DECODER PROJECTION

Figure 1 reveals that the SAE decoder latents contain mixes of underlying features, both when the L0 is too high and also when it is too low. As the SAE approaches the correct L0, each SAE latent has fewer components of multiple true features mixed in, becoming more monosemantic. Thus, we expect that when the SAE is at the correct L0, most latents should have near zero projection on arbitrary training inputs, because they usually do not contain the feature being tracked by that latent. If we are far from the correct L0, then SAE latents contain components of many underlying features, and we expect latents to project more strongly on arbitrary training inputs.

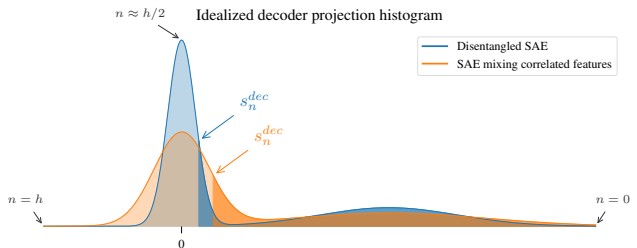

Figure 17: Idealized histogram of decoder projections on input activations demonstrating the intuition behind our $n^{\text{th}}$ decoder projection metric, $s_n^{\text{dec}}$. For an arbitrary input, most latents should be non-active and thus have low projection. When SAE latents are monosemantic, meaning they do not mix components of many features, we expect non-active latents to have a near-zero projection on arbitrary input activations. However, if SAE latents mix positive and negative components of many underlying features, then those latents will have larger projections on arbitrary inputs that contains those features. By picking an $N$ less than $h/2$ (corresponding roughly to the origin), a smaller $s_n^{\text{dec}}$ means latents have smaller projection on arbitrary inputs and thus are more monosemantic.

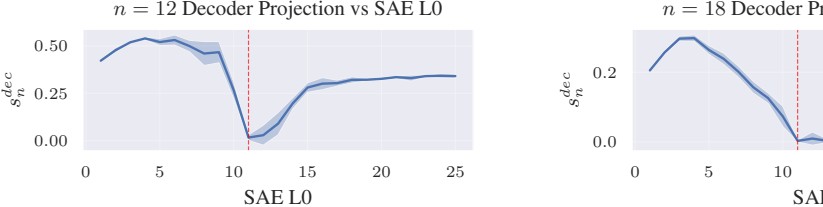

Figure 18: $n^{\text{th}}$ decoder projection vs SAE L0 for $n = 12$ (left) and $n = 18$ (right) on our toy model SAEs. The true L0, 11, is marked by a dotted line on the plots. Both settings of $n$ are minimized at the true L0, but the slopes of the metric change depending on N. The shaded area is 1 stdev.

We now define a metric we call $n^{th}$ *decoder projection score*, or $s_n^{\text{dec}}$, that we can use to find the optimal L0 of the SAE. Given SAE inputs $\mathbf{x} \in \mathbb{R}^{b \times d}$ where $b$ is the batch size and $d$ is the input dimension, we first compute the decoder projections for all latents:

$$\mathbf{Z} = (\mathbf{x} - \mathbf{b}_{\text{dec}})\mathbf{W}_{\text{dec}}^{\top} \in \mathbb{R}^{b \times h} \tag{58}$$

where $\mathbf{b}_{\text{dec}} \in \mathbb{R}^d$ is the decoder bias and $\mathbf{W}_{\text{dec}} \in \mathbb{R}^{d \times h}$ is the decoder weight matrix with $h$ latent dimensions. To aggregate across the batch, we flatten $\mathbf{Z}$ to obtain $\mathbf{z} \in \mathbb{R}^{bh}$ and sort these values in descending order to get $\mathbf{z}_{\downarrow}$. The $n^{\text{th}}$ decoder projection is then defined as:

$$s_n^{\text{dec}} = \mathbf{z}_{\downarrow}[n \cdot b] \tag{59}$$

where $[n \cdot b]$ corresponds to selecting the element at index $n \cdot b$. The multiplication by $b$ accounts for the batch dimension, effectively selecting the $n^{\text{th}}$ highest projection value when considering all samples in the batch. For this to work $n$ should be sufficiently larger than a reasonable guess at the correct L0, as in a perfect SAE, the decoder for these latents should be uncorrelated with input activations. Picking any $n$ up to $h/2$ should work, as the majority of latents should have low projection on arbitrary input activations, so $h/2$ intuitively corresponds to 0 expected projection. The intuition behind $s_n^{dec}$ is shown visually in Figure 17, and is formalized in Appendix A.10.

We calculate $s_n^{\text{dec}}$ for $n = 12$ and $n = 18$, varying SAE L0 from 2 to 25 with 5 seeds per L0 in Figure 18. The metric is minimized at the true L0, 11, in both cases, although the shape changes depending on $n$. In both cases, the slope of $s_n^{\text{dec}}$ is flat when L0 is slightly higher than the true L0.

### A.9.1 LLM SAE RESULTS

Next, we $s_n^{\text{dec}}$ for each LLM SAE we evaluated in the paper along with k=16 sparse probing results. Gemma-2-2b layer 5 and Llama-3.2-1b layer 7 results are shown in Figure 19. The results roughly match what we saw with $c_{\text{dec}}$.

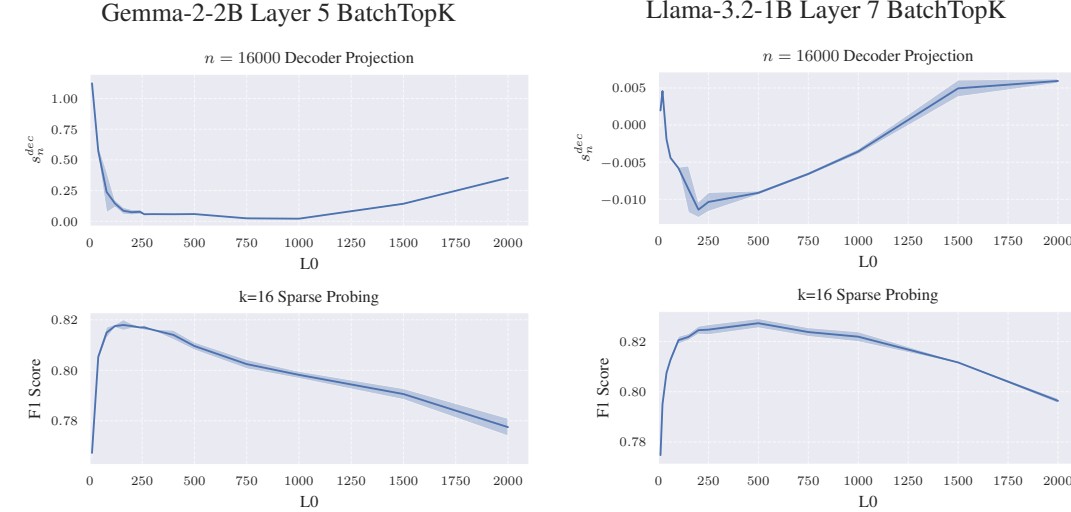

Figure 19: $n^{th}$ decoder projection and k=16 sparse probing results for BatchTopK SAEs trained on Gemma-2-2b layer 5 (left) and Llama-3.2-1b layer 7 (right). The metric is roughly minimized near peak sparse-probing performance. The shaded area is 1 stdev.

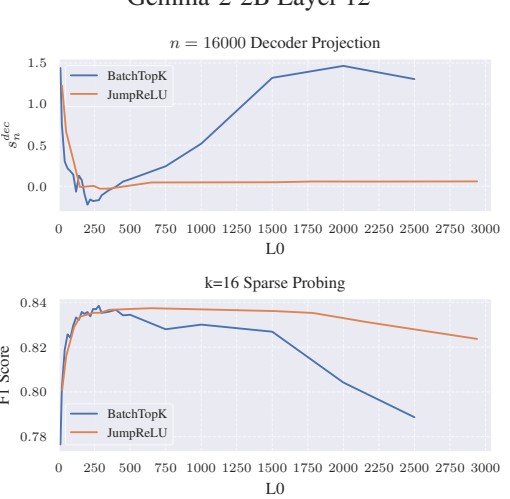

Figure 20: $n^{th}$ decoder projection and sparse probing results for BatchTopK and JumpReLU SAEs trained on Gemma-2-2b layer 12. The metric seems to align less well with k=16 sparse probing results.

BatchTopK and JumpReLU results for Gemma-2-2b layer 12 are shown in Figure 20. The results look similar to what we saw for $c_{dec}$.

### A.9.2 WHICH METRIC IS BETTER?

We choose to focus on $c_{dec}$ as it is a simpler metric both to understand and implement as it has no hyperparameters. However, we expect that when an SAE is near the correct L0, there are likely many indicators that all should point to similar results. Any metric which can detect correlated features being mixed into SAE latents should give roughly similar results.

## A.10 THEORETICAL JUSTIFICATION FOR $s_n^{\text{DEC}}$ METRIC

We provide a theoretical justification for why the $n^{\text{th}}$ decoder projection metric, $s_n^{\text{dec}}$, successfully identifies when SAE latents are mixing correlated features.

**Theorem 3.** *Consider two SAEs with identical dictionary sizes $h$ and input dimension $d$, where SAE 1 has greater feature mixing than SAE 2. Specifically, let the decoder projections onto a feature $f$ for non-active latents follow:*

$$\text{SAE 1:} \quad z_i^{(1)} \sim \mathcal{N}(0, \sigma_0^2 + \sigma_1^2) \tag{60}$$

$$\text{SAE 2:} \quad z_i^{(2)} \sim \mathcal{N}(0, \sigma_0^2 + \sigma_2^2) \tag{61}$$

*where $\sigma_0^2$ represents the base variance from superposition noise, and $\sigma_1^2 > \sigma_2^2$ represents the variance from feature mixing. Then for $n < h/2$, we have $\mathbb{E}[s_n^{dec}]$ is larger for SAE 1 than SAE 2.*

*Proof.* Let $f \in \mathbb{R}^d$ be an underlying true feature with $\|f\|_2 = 1$. Consider an SAE with decoder $\mathbf{W}_{\text{dec}} \in \mathbb{R}^{d \times h}$ and decoder bias $\mathbf{b}_{\text{dec}} \in \mathbb{R}^d$. For an input activation $\mathbf{x} \in \mathbb{R}^d$, the projection of latent $i$ onto the input is:

$$z_i = (\mathbf{x} - \mathbf{b}_{\text{dec}})^\top \mathbf{W}_{\text{dec},i} \tag{62}$$

**Decomposition of decoder latents.** We decompose each decoder latent $\mathbf{W}_{\text{dec},i}$ into three components:

$$\mathbf{W}_{\text{dec},i} = \alpha_i f + \beta_i g_i + \epsilon_i \tag{63}$$

where:

- $\alpha_i f$ is the component aligned with feature $f$ (the intended feature for latent $i$ if $i$ is the correct latent, or mixing if $i$ is incorrect)

- $\beta_i g_i$ represents components of other correlated/anti-correlated features mixed into latent $i$, where $g_i$ is orthogonal to $f$

- $\epsilon_i$ represents superposition noise, also orthogonal to $f$

**Distribution of projections for non-active latents.** Consider latents that should not activate for feature $f$ (i.e., latents $i$ where $\alpha_i$ should ideally be near zero). For an input $\mathbf{x}$ containing feature $f$ with magnitude $m_f$, we can write:

$$\mathbf{x} - \mathbf{b}_{\text{dec}} = m_f f + \mathbf{r} \tag{64}$$

where $\mathbf{r}$ contains all other feature contributions orthogonal to $f$.

The projection of latent $i$ becomes:

$$z_i = (m_f f + \mathbf{r})^\top (\alpha_i f + \beta_i g_i + \epsilon_i) = m_f \alpha_i + \mathbf{r}^\top (\beta_i g_i + \epsilon_i) \tag{65}$$

For non-active latents in a well-trained SAE, we expect:

- $\alpha_i \approx 0$ for the intended feature component

- $\beta_i g_i$ represents unintended mixing of correlated features

- $\epsilon_i$ represents superposition noise

**Modeling as Gaussian mixtures.** Under the assumptions that:

1. Feature magnitudes $m_f$ and residual components $\mathbf{r}$ vary across the input distribution

2. The number of latents $h$ is large

3. Feature mixing coefficients $\beta_i$ arise from optimization pressure to compensate for insufficient L0

By the Central Limit Theorem, the distribution of projections $z_i$ for non-active latents approximately follows:

$$z_i \sim \mathcal{N}(0, \sigma_{\text{base}}^2 + \sigma_{\text{mix}}^2) \tag{66}$$

where:

- $\sigma_{\text{base}}^2$ captures variance from superposition noise ($\epsilon_i$)

- $\sigma_{\text{mix}}^2$ captures variance from feature mixing ($\beta_i g_i$)

**Comparing two SAEs.**   Consider two SAEs:

- **SAE 1** (high feature mixing): $z_i^{(1)} \sim \mathcal{N}(0, \sigma_{\text{base}}^2 + \sigma_1^2)$ where $\sigma_1^2$ is large

- **SAE 2** (low feature mixing): $z_i^{(2)} \sim \mathcal{N}(0, \sigma_{\text{base}}^2 + \sigma_2^2)$ where $\sigma_2^2$ is small

with $\sigma_1^2 > \sigma_2^2$.

**Computing $s_n^{\text{dec}}$.**   For a batch of size $b$, we have $bh$ projection values. After sorting in descending order, the $n^{\text{th}}$ decoder projection is:

$$s_n^{\text{dec}} = z_\downarrow[n \cdot b] \tag{67}$$

This corresponds to the $(n \cdot b)/(bh) = n/h$ quantile of the distribution. For $n < h/2$, this is the $(n/h)^{\text{th}}$ quantile on the positive side of the distribution.

**Quantile comparison.**   For a standard normal distribution $Z \sim \mathcal{N}(0, 1)$ and $\sigma_1 > \sigma_2 > 0$, the $p^{\text{th}}$ quantile satisfies:

$$Q_p(\mathcal{N}(0, \sigma_1^2)) = \sigma_1 \cdot Q_p(\mathcal{N}(0, 1)) > \sigma_2 \cdot Q_p(\mathcal{N}(0, 1)) = Q_p(\mathcal{N}(0, \sigma_2^2)) \tag{68}$$

for $p > 0.5$ (corresponding to positive quantiles).

Since $n < h/2$ implies $p = n/h < 0.5$, we are actually looking at the $(1 - p)^{\text{th}}$ quantile on the right tail due to descending sort. This gives us:

$$\mathbb{E}[s_n^{\text{dec}}]_{\text{SAE 1}} > \mathbb{E}[s_n^{\text{dec}}]_{\text{SAE 2}} \tag{69}$$

Therefore, SAEs with greater feature mixing (larger $\sigma_{\text{mix}}^2$) will have larger values of $s_n^{\text{dec}}$, justifying its use as a metric for detecting feature mixing. □

**Remark 2.** *The choice of $n \approx h/2$ in practice corresponds to sampling from a region where the distribution is sensitive to changes in variance (roughly near the median), while being sufficiently far from the extreme tails to maintain statistical stability. Values of $n$ too close to 0 would sample from the extreme right tail where variance is high, while $n$ too close to $h$ would sample from regions dominated by active latents rather than the non-active latents we wish to characterize.*

**Remark 3.** *This theoretical analysis assumes that decoder projections follow approximately Gaussian distributions. While this is a simplification, our empirical results in both toy models (where we have full control) and LLM SAEs support this assumption, as evidenced by the decoder projection histograms in Figure 9.*

### A.11   Automatically finding the correct L0 during training

A natural next step of our finding that the correct L0 occurs when Nth decoder projection, $s_n^{\text{dec}}$, metric is minimized is to use this to find the correct L0 automatically during training. This is a meta-learning task, as the L0 is a hyperparameter of the training process. We find there are several challenges to directly using $s_n^{\text{dec}}$ as an optimization target:

- **Small gradients directly above correct L0** In our plots of $s_n^{\text{dec}}$ from both toy models and Gemma-2-2b, we find that the metric is relatively flat in a region start at the correct L0 and extending to higher L0 values. We thus need a way to traverse this flat region and stop once the metric starts to increase again.

- **The impact of changing L0 is delayed** We find that it takes many steps after changing L0 for $s_n^{\text{dec}}$ to also change, meaning it is easy to overshoot the target L0 or oscillate back and forth.

- **Dropping L0 too low can harm the SAE** As we saw in Appendix A.8, if the L0 is too low the SAE can permanently end up in poor local minima. We thus want to avoid dropping below the correct L0, even temporarily, to avoid permanently breaking the SAE. We therefore need to start with L0 too high and slowly decrease it until we find the correct L0.

- **Noise during training** We find that while $s_n^{\text{dec}}$ shows clear trends after training for many steps, it can be noisy on each training sample. So our optimization needs to be robust to this noise.

Taking these requirements into account, we present an optimization procedure to find the L0 that minimizes $s_n^{\text{dec}}$ automatically during training. We first estimate the gradient of $s_n^{\text{dec}}$, hereafter referred to as to as the metric, $m$, with respect to L0, $dm/dL0$. We first define an evaluation step $t$ as a set number of training steps (we evaluate every 100 training steps). At $t$ we change L0 by $\delta_{L_0}$. At the next evaluation step, $t+1$, we evaluate $m$. We use a sliding average of $s_n^{\text{dec}}$ over the past 10 training steps to calculate $m$ to help account for noise. We the estimate $dm/dL0$ as:

$$\frac{dm}{dL0} = \frac{m_{t+1} - m_t}{\delta_{L0}}$$

Next, we add a small negative bias to this gradient estimate to encourage our estimate to push L0 lower even if the loss landscape is relatively flat. We use a bias magnitude $0 < b < 1$ that is multiplied by the magnitude of our gradient estimate, so that our biased estimate can never change the sign of the gradient estimate, but can gently nudge it to be more negative in flat, noisy regions of the loss landscape. We find $b = 0.1$ works well. Thus, our biased gradient estimate $dm_b/dL0$ is calculated as below:

$$\frac{dm_b}{dL0} = \frac{dm}{dL0} - b \left| \frac{dm}{dL0} \right|$$

We then provide this gradient to the Adam optimizer (Kingma & Ba, 2014) with default settings, and allow it to change the L0 parameter.

We add the following optional modifications to this algorithm. First, we clip the gradient estimates $dm/dL0$ to be between -1 and 1. We also set a minimum and maximum $\delta_{L_0}$. The minimum is added to avoid the denominator of our gradient estimate being near 0, and the maximum is chosen to keep the L0 from changing too quickly. In practice, we find a minimum $\delta_{L_0}$ between 0 and 1 seems to work well, and a maximum $\delta_{L_0}$ between 1 and 5 seems to work well.

We find that this optimization strategy works very well in toy models, but requires a lot of hyper-parameter tuning to work in real LLMs, limiting its utility. The starting L0, $n$ for $s_n^{\text{dec}}$, $b$, learning rate for the Adam optimizer, and min and max $\delta_{L_0}$ values all have a big impact on how fast and how aggressively the optimization works. The slope of $m$ around the correct L0 is shallow, so it is easy to overshoot. We also find that different values of $n$ take more or less time to converge during training. We expect it is possible to further simplify and improve this process in future work.

## A.12 EXTENDED LLM RESULTS

We include further results for Gemma-2-2b layer 20, to extend the analysis to later model layers. Results are shown in Figure 21.

## A.13 L0 OF OPEN-SOURCE SAES ON NEURONPEDIA

We analyze common open-source SAEs as provided by Neuronpedia (Lin, 2023) and SAELens (Bloom et al., 2024). We include all SAEs cross-listed in both SAELens and Neuronpedia with an L0 reported in SAELens. We show the results as a histogram in Figure 22. Our analysis shows that for layer 12 of Gemma-2-2b, the correct L0 should be around 200-250. However, we find that most open-source SAEs have L0 below 100, much lower than our analysis expects to be ideal.

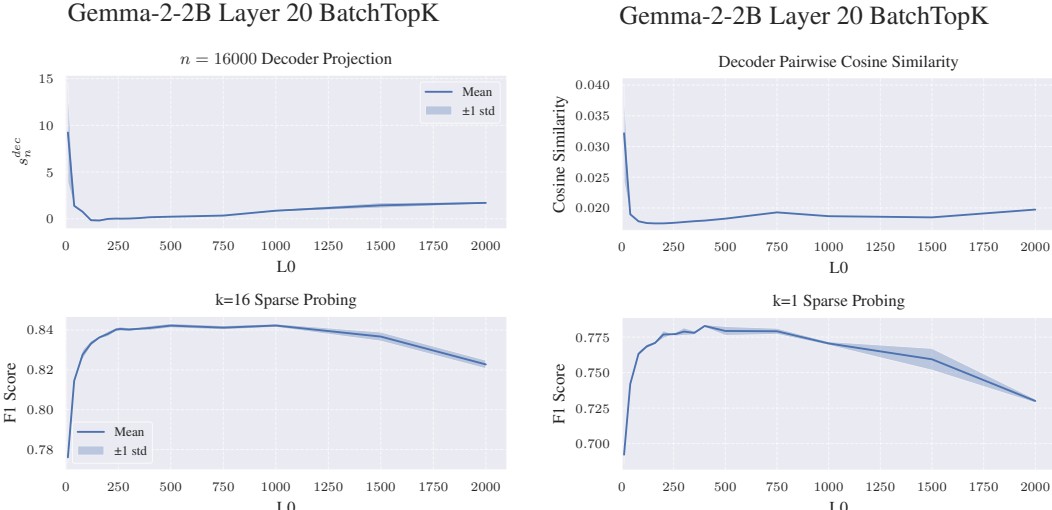

Figure 21: $N^{th}$ decoder projection (top left) and decoder pairwise cosine similarity (top right) with K=16 sparse probing results (bottom-left) and K=1 sparse probing results (bottom-right) for Batch-TopK SAEs trained on Gemma-2-2b layer 20.

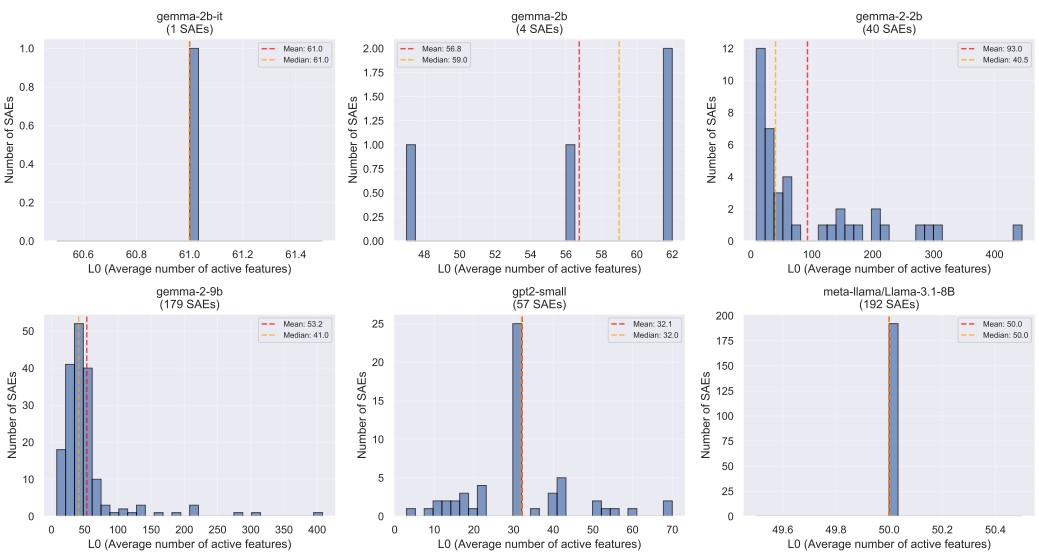

Figure 22: L0 of SAEs on Neuronpedia with known L0 listed in SAELens.

## A.14  LIMITATIONS

We limited the scope of our investigation to features satisfying the linear representation hypothesis, and do not investigate how SAEs react if the underlying features are actually non-linear (Engels et al., 2025). However, we do not feel that non-linear features are necessary for SAEs to fail to work properly, as we demonstrate in this paper. We also do not consider the nuances of how unbalanced correlations impact the SAE, as simple correlations are already enough to cause problems. However, we do expect that different sorts of correlations may affect SAEs differently, and would encourage future work to look into this. Finally, we only investigated a few layers of popular LLMs, as running sweeps of SAE training at every layer of the LLM was too prohibitively expensive for this work. Nevertheless, we have no reason to expect any meaningfully different behavior in decoder projection at other LLM layers.

### A.15 EXTENDED NTH DECODER PROJECTION PLOTS

In this section we document $n^{\text{th}}$ decoder projection plots for multiple values of N for each sweep of L0 we performed. We show Llama-3.2-1b layer 7 plots in Figure 25, Gemma-2-2b layer 5 in Figure 23, and Gemma-2-2b layer 12 in Figure 24. We note that in all cases, low L0 behavior is similar: no matter the value of N, $s_n^{\text{dec}}$ increases dramatically at low L0. However, the high L0 behavior is less consistent. We always see a similar "elbow" in the plots at roughly the same place regardless of N, but sometimes this elbow corresponds to a clear global minimum, and sometimes the high L0 behavior is very shallow. We find that using a N near $h/2$ ( 16k in our cases) seems to give the best results.

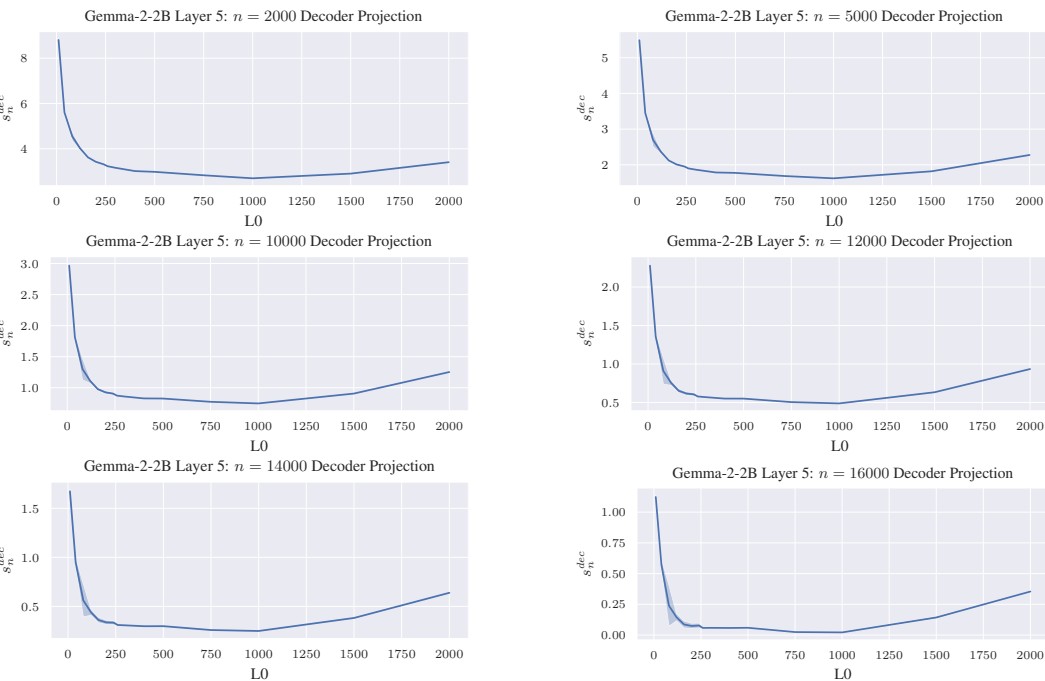

Figure 23: Extended Nth decoder projection plots. Gemma-2-2b, layer 5, 32k latents. These plots never have a clear global minimum at the "elbow" point, but the "elbow" is always at the same point regardless of choice of N.

### A.16 EXTENDED ANALYSIS OF JUMPRELU VS BATCHTOPK DYNAMICS

JumpReLU and BatchTopK SAEs are both considered state of the art, but we find they have notable differences in their behavior at high L0 in our experiments. In this section, we explore what maybe be causing these differences. In theory, JumpReLU and BatchTopK SAEs are very similar, as a BatchTopK SAE can be viewed as a JumpReLU SAE with a single global threshold, rather than a threshold per-latent (Bussmann et al., 2024). However, the training losses are quite different for JumpReLU vs BatchTopK. We use the JumpReLU variant laid out by Conerly et al. (2025), which allows gradients to flow through the JumpReLU threshold to the rest of the model parameters. We expect this means that JumpReLU SAEs are better able to coordinate the threshold with the rest of the model parameters, while BatchTopK cannot, as the threshold does not directly receive a gradient in BatchTopK training.

We begin by comparing the encoder bias between JumpReLU and BatchTopK in Figure 26. We see that BatchTopK SAEs rely much more heavily on the encoder bias than JumpReLU SAEs seem to, with a much wider variance in values and a sharper decrease compared to JumpReLU. We expect this is because BatchTopK cannot coordinate the cutoff threshold with the encoder directly as JumpReLU can, since there is no gradient available to directly change the threshold of BatchTopK SAEs.

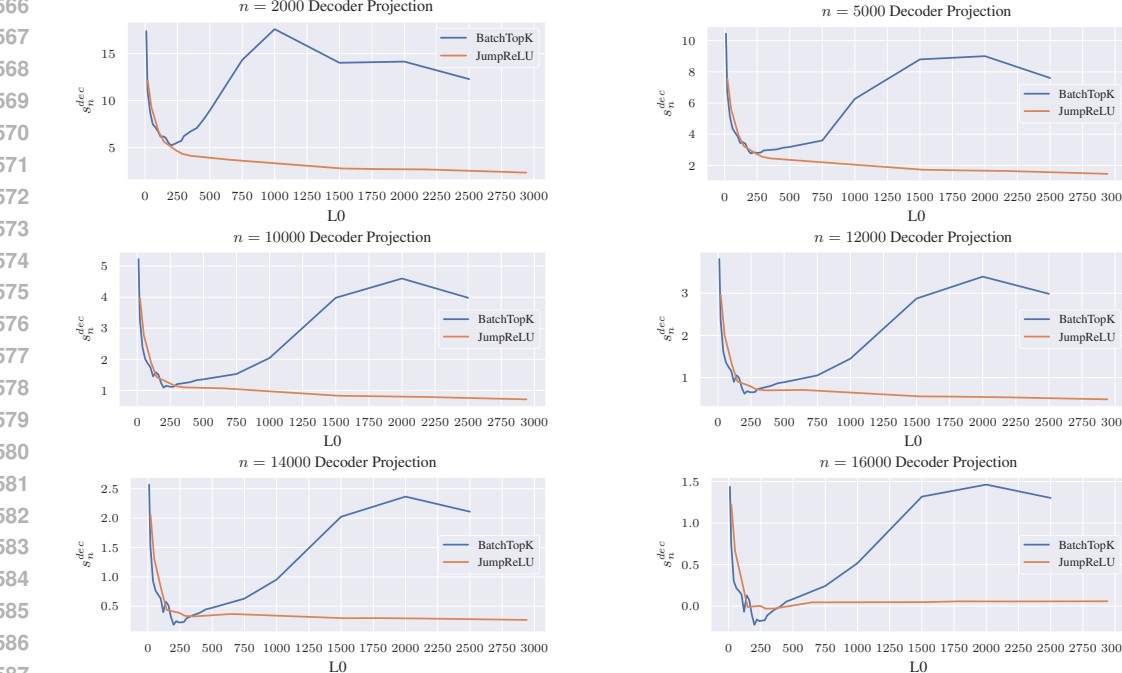

Figure 24: Extended Nth decoder projection plots. Gemma-2-2b, layer 12, 32k latents for both JumpReLU and BatchTopK. For BatchTopK, regardless of the choice of N, all plots are minimized around the same L0 range, 200-250. For JumpReLU, there is a clear "elbow" at roughly the same L0, but this elbow is only a clear minimum at N=16k.

Next, we inspect the threshold values between JumpReLU and BatchTopK in Figure 27. Here as well, we see dramatic differences between BatchTopK and JumpReLU SAEs. The threshold for BatchTopK is much higher than it is for JumpReLU, and the threshold decreases as L0 increses. This makes sense, since using a lower cutoff means more latents can fire. However, JumpReLU seems to unintuitively have the opposite trend, with the threshold actually *increasing* with L0. We saw in Figure 26 that the encoder bias for JumpReLU (and BatchTopK) SAEs increases as well as L0 increases, so perhaps this increase in threshold for JumpReLU SAEs with increasing L0 is just to offset that trend somewhat. We also notice that the variance in JumpReLU SAE thresholds also increases as L0 increases, supporting our hypothesis that one of the reasons JumpReLU SAEs seem to handle high L0 better than BatchTopK is because the thresholds are able to dynamically adjust to near the correct cutoff point per latent, aleviating the situation we saw in BatchTopK SAEs where we can be at both too high and too low L0 at the same time (Section 4.2).

## A.17 PYTORCH PSEUDOCODE FOR METRICS

We present Pytorch pseudocode for nth decoder projection in Figure 29 and decoder pairwise cosine similarity in Figure 28.

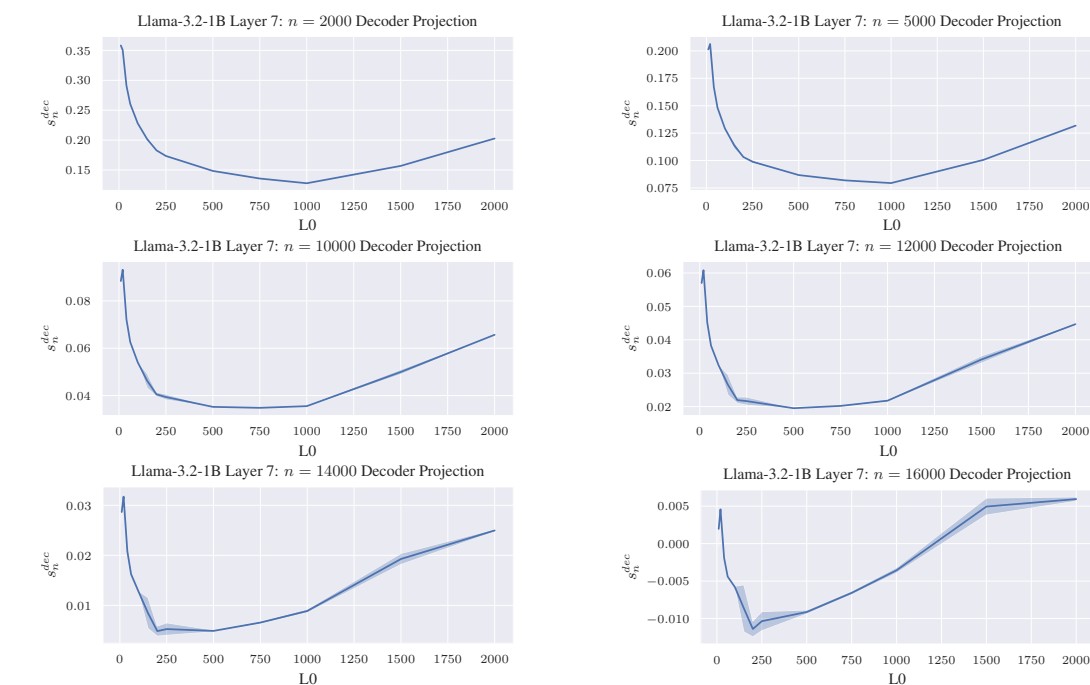

Figure 25: Extended Nth decoder projection plots. Llama-3.2-1b, layer 7, 32k latents. The plots begin to have a sharp minimum around N=14k, but the "elbow" of the plots before the decoder projection increases at low L0 is always around the same location.

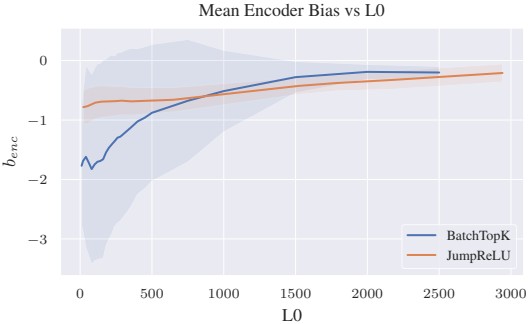

Figure 26: Mean encoder bias vs L0. Shaded area in plots corresponds to 1 stdev.

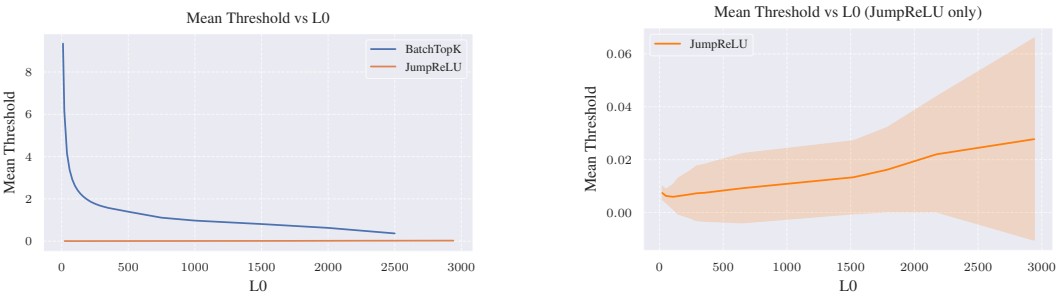

Figure 27: Threshold vs L0 for JumpReLU and BatchTopK SAEs. Shaded area in plots corresponds to 1 stdev. Interestingly, JumpReLU threshold is much lower than the BatchTopK threshold, and actually increases as L0 increases. We plot just JumpReLU on its own (right) since it is otherwise difficult to see these trends, as the threshold is so much smaller than BatchTopK.

```
def pairwise_decoder_cosine_similarity(sae):
    norm_dec = torch.nn.functional.normalize(sae.W_dec, dim=1)
    dec_sims = torch.mm(norm_dec, norm_dec.T)
    triu_mask = torch.triu(
        torch.ones_like(dec_sims),
        diagonal=1,
    ).bool()
    return dec_sims[triu_mask].abs().mean()
```

Figure 28: Pytorch pseudocode for decoder pairwise cosine similarity

```
def nth_decoder_projection(input_acts, sae, n):
    dec_proj = (input_acts - sae.b_dec) @ sae.W_dec.T
    sorted_dec_proj = dec_proj.flatten().sort(descending=True)
    index = n * dec_proj.shape[0]
    return sorted_dec_proj.values[index]
```

Figure 29: Pytorch pseudocode for $n^{th}$ decoder projection

