# OpenReview forum: "Sparse but Wrong: Incorrect L0 Leads to Incorrect Features in Sparse Autoencoders"
_ICLR.cc/2026/Conference — Submitted to ICLR 2026_

### Official Review · Reviewer_cz9D · 2025-10-23

**Soundness:** 3
**Presentation:** 3
**Contribution:** 4
**Rating:** 8
**Confidence:** 4

**Summary:**

The author of this paper investigate how the sparsity of a Sparse Autoencoder influences the learned features by the SAE. The authors show in toy models how using a L0 lower than the real L0 can lead to a bad set of learned features, similarly to what happens if the SAE is trained with a very high L0. They show that in toy settings SAEs with lower than the true L0 can 'cheat' and achieve better reconstruction loss. With this in mind they propose metrics to measure if SAEs were trained with too low L0 without know the ground truth labels. They apply this method to LLMs and find that the best L0 for probing is also maximizes their metric.

**Strengths:**

Understanding what is the correct L0 to use is a very important problem, for the comunity.
Their metric seems to capture the real L0 on toy models.
This work presents convincing evidence that researchers might have been using too low of L0 when analysing LLMs.

**Weaknesses:**

The presentation of the paper is sightly subpar. Figures could be improved.
The paper does not discuss a relevant work 'Interpretability as Compression: Reconsidering SAE: Explanations of Neural Activations with MDL-SAEs'

**Questions:**

What are the reconstruction values for the SAEs of Figure 5? The caption is a bit confusing. Are these related to the ones shown in Figure 4? Maybe these figures chould be shown together?
What do the authors think is the relationship, if any, between you proposed metrics of finding the optimal L0 and the ones described in 'Interpretability as Compression: Reconsidering SAE Explanations of Neural Activations with MDL-SAEs'?
What is the intuition for selecting the different values of n?

---

> ### Author Response · Authors · 2025-11-20
>
> We thank the reviewer for their thoughtful reading of our work and kind words about the strengths of the paper. We address the follow-up questions below:
>
> > The paper does not discuss a relevant work 'Interpretability as Compression: Reconsidering SAE: Explanations of Neural Activations with MDL-SAEs'
>
> > What do the authors think is the relationship, if any, between you proposed metrics of finding the optimal L0 and the ones described in 'Interpretability as Compression: Reconsidering SAE Explanations of Neural Activations with MDL-SAEs'?
>
> Thank you for pointing out this connection We have added a discussion of MDL-SAEs to the related work section of the paper. The MDL-SAEs work takes the perspective that there may not be such a thing as a correct setting for SAE width and L0, as sparsity-reconstruction trade-off plots would push us towards an infinitely wide SAE with L0=1, which is clearly nonsensical. To address this, they suggest that information theory may provide a good metric by which to select these hyperparameters.
>
> Our work comes from the opposite perspective, asking if LLMs do in-fact have linear features that follow the linear representation hypothesis (LRH), under what cases can we trust an SAE will find them? We then show that if L0 is not set correctly the SAE will not find underlying features if they exist, and that instead the SAE will “cheat” and mix correlated features together in its latents.
>
> One important point of agreement between our work and the MDL-SAEs work, though, is that sparsity-reconstruction tradeoff plots are not a good way to evaluate SAE architectures.
>
> > What are the reconstruction values for the SAEs of Figure 5? The caption is a bit confusing. Are these related to the ones shown in Figure 4? Maybe these figures chould be shown together?
>
> Thank you for the suggestion, we have now moved the figures next to each other and made it clear that the SAEs from Figure 5 are simply datapoints from Figure 4.

---

> > ### Comment · Reviewer_cz9D · 2025-11-23
> >
> > I want to begin my comment by disagreeing with some of the statements from reviewer Z23o. The comment that 'It has been a well-established fact since the earliest days of machine learning that poor hyperparameter selection leads to underperforming models. It is not clear to me that the paper makes any contribution beyond showing that this is also true of the L0 hyperparameter when training SAEs.' seems to either ignore all the SAE literature up until today or to ignore it. It also seems to ignore the multitude of literature that exists in machine learning research indicating how to better chose hyperparameters.
> >
> > Having read other reviewers comments and the author's answers, I will keep my score

---

> > > ### Author Response · Authors · 2025-11-30
> > >
> > > We thank the reviewer for their continued engagement and for maintaining their strong score. We further appreciate the support regarding Reviewer Z23o’s review. We hope this helps further clarify that dismissing the L0 problem as simply "bad hyperparameter selection" overlooks the context of current SAE literature, where L0 is not treated as just a standard hyperparameter to be optimized.

---

### Official Review · Reviewer_mo6z · 2025-10-31

**Soundness:** 3
**Presentation:** 3
**Contribution:** 3
**Rating:** 4
**Confidence:** 3

**Summary:**

Authors state that most SAE users choose the wrong sparsity, impeding the recovery of true model features. This stems from over-reliance on sparsity-reconstruction tradeoff plots during SAE parameter search, most often causing users to end up with SAE L0 that is too low. The authors explore the consequences of SAEs with low L0 relative to true L0 using toy models and find that low L0 causes 1) imperfect recovery of real features but 2) low reconstruction error. They address the mechanism, finding that low L0 SAEs mix features based on feature correlation statistics to achieve better variance explained than ground-truth SAE features. Next, they develop metrics to identify if L0 is less than true L0 in toy models. The core results are extended to real LLMs, and while true L0 is unkown for these systems, the authors’ metrics capture the L0 that results in the best sparse probing performance for LLMs.  Finally, the authors explore the differences between two SOTA SAE architectures, finding that JumpReLU SAEs are more robust to high L0 than BatchTopK SAEs, with both architectures performing similarity at lower than ‘true’ L0. High L0 is also addressed, but results are more mixed.

**Strengths:**

The authors demonstrate a key limitation of standard parameter selection practices used by SAE practitioners: that SAEs with low L0 are often selected based on sparsity-reconstruction tradeoff analyses yet impede true feature recovery.

They develop two metrics, both capturing when L0 is too low in toy models, validating the results on real data from multiple LLMs.

They test their metrics on two SOTA SAE architectures.

The result is a timely, well-tested toolkit for SAE evaluation that can be adopted widely.

**Weaknesses:**

One thing that I really miss in this paper: You are saying that rate-distortion heuristics (like picking the lowest L0 with the lowest MSE) does not give the most disentangled SAE. Instead you are proposing a different heuristic. If that heuristic is better, then you should be able to just take a bunch of pretrained SAEs from SAEbench and show how your heuristic leads to a better SAE across the different metrics. What is stopping you from doing that experiment?

Please include references and look for overlap with existing literature on estimating the number of sources in ICA (https://www.nature.com/articles/s41598-023-49355-z, https://pmc.ncbi.nlm.nih.gov/articles/PMC6871474/)

Presentation consistency. 1) Would like decoder projection plots for all model types shown 2) Plot high L0 alongside low L0 throughout the main text, particularly since it differentiates the two SAE architectures at some point.

Selection of n: Though explained, selection of n seems somewhat arbitrary at times. Would be more compelling to show the average across n results, fix the main text n value to be h/2 consistently, or show a few n values flanking and including the h/2 value. Relatedly, would add h value to fig caption where n is selected for clarity.

Alternate explanations: Feature frequency could also explain aspects of particularly low L0 performance. The left vs right side of Fig 1 (left) hints at this. Plotting feature recovery as a function of feature frequency would be helpful.

**Questions:**

- Does the degree of correlation influence the degree of mixing?

- Is there more evidence of feature mixing (/absorption) in the low L0 setting but feature splitting in the high L0 setting?

- Can you probe interaction effects (frequency and correlation?) for example, by replicating figs 2, 3 where f0 has relatively high and low frequency?

- In cases where h/2 is not the ‘best’ n, can you add to your metric (ie with a quantitative measure to capture your finding in lower right panel of Fig 10)? If not always, then how often does it work, and in what settings?

- Add plot(s) and/or table(s) showing the SAE resulting from elbow method vs your metric(s)

---

> ### Author Response · Authors · 2025-11-20
>
> We thank the reviewer for their thoughtful reading of our work. We address the questions below:
>
> > You are saying that rate-distortion heuristics (the lowest L0 with lowest MSE) does not give the most disentangled SAE.
>
> We clarify that current literature treats L0 not as a parameter to be optimized, but as a user preference for reconstruction quality (hence the Pareto curves). There is no single point that has both lowest L0 and lowest MSE, since lowering L0 increases MSE, and thus why it is a trade-off. Sparsity-reconstruction plots are not for choosing an L0, but for evaluating an SAE architecture at multiple L0s, implying all L0s are equally vaild.
>
> We are asserting the opposite: there is a (mostly) correct L0 for a given model and data distribution, and we should strive to find it. Thus, we are not comparing our metric for finding the correct L0 to sparsity-reconstruction trade-off plots for finding the correct L0, because sparsity-reconstruction trade-off plots are not a method for finding the correct L0.
>
> Furthermore, we show the idea of a “sparsity-reconstruction tradeoff”, despite being ubiquitous in the literature, is a flawed way to evaluate SAE architectures. We show in S.3.4 that an incorrect SAE that mixes correlated features scores better than the ground-truth correct SAE, despite learning corrupted, incorrect latents. You can view this as a proof by counterexample, disproving the notion that “sparsity-reconstruction tradeoff” plots are a valid way to evaluate SAE architectures.
>
> If we misunderstood your question, please let us know and we will follow-up!
>
> > … pretrained SAEs from SAEbench and show how your heuristic leads to a better SAE across the different metrics
>
> Indeed, one of the main motivations for our work was noting that recent work like SAEBench shows that low-L0 SAEs perform very poorly, and we wanted to understand why that is. In our work, we compare our metric against sparse probing performance (a proxy for 'better SAEs') in Figures 8 and 9, showing our metric correlates with downstream utility while MSE-based heuristics do not.
>
> > Please include references ... in ICA
>
> Thank you for sharing these works from the ICA field, we have included them in related work.
>
> > Does the degree of correlation influence the degree of mixing?
>
> Yes, absolutely! The higher the correlation the more mixing we see. Figures 2 and 3 show positive and negative correlations cause positive and negative mixing, but any intermediate value also causes an intermediate amount of mixing (A.3.1). If there is no correlation, then we see no mixing (latents 2 - 4 in those show no mixing with each other due to having no correlation with each other). We added further results demonstrating the effect of correlation strength in A.3.1.
>
> > Is there more evidence of feature mixing (/absorption) in the low L0 setting but feature splitting in the high L0 setting?
>
> In the low L0 setting, there is gradient pressure that forces mixing into SAE latents to improve reconstruction loss. At the high L0 setting, there is not this gradient pressure but there are many ways to reconstruct the input that do not result in learning correct features (and thus it looks like latents have mixtures of underlying features). Our understanding is that feature splitting is due to increasing the SAE width rather than L0 [1].
>
> > Can you probe interaction effects (frequency and correlation?)
>
> We include frequency-based interactions in the large toy model section (Section 3.2), by linearly decreasing feature firing frequency with feature index. So, lower feature indices fire more frequently than higher indices. We do see clear effects from this in the results. Interestingly, latents tracking higher frequency features are more adversely affected by too low L0, while latents tracking lower frequency features are more adversely affected by too high L0.
>
> > In cases where h/2 is not the ‘best’ n, can you add to your metric … how often does it work, and in what settings?
>
> There is not necessarily a “best” n, nearly any value of n results in the same “elbow” point in the plot. We demonstrate this with multiple values of n in Figures 23, 24, and 25 in the Appendix.
>
> We also swapped the $s_n^{dec}$ metric in the main body with decoder pairwise cosine similarity, $c_{dec}$, previously the “alternative metric” in the manuscript. Several reviewers found the selection of $n$ confusing, and $c_{dec}$ avoids this issue as it has no hyperparameters while similarly tracking correlations in the decoder and yielding similar results.
>
> > Add plot(s) and/or table(s) showing the SAE resulting from elbow method vs your metric(s)
>
> We include plots of our metric and sparse probing scores in Figures 9 and 10
>
> ### References
> [1] Bricken, T., Templeton, A., Batson, J., Chen, B., Jermyn, A., Conerly, T., Turner, N., Anil, C., Denison, C., Askell, A., et al. Towards monosemanticity: Decomposing language models with dictionary learning. Transformer Circuits Thread, 2, 2023. URL

---

### Official Review · Reviewer_Pdhb · 2025-10-31

**Soundness:** 2
**Presentation:** 3
**Contribution:** 3
**Rating:** 6
**Confidence:** 3

**Summary:**

The paper investigates the problem of incorrect sparsity target (L0) in Sparse Autoencoders
(SAEs). It demonstrates that when L0 is set too low or too high, learned latents mix correlated
and anti-correlated features, losing monosemanticity. Through controlled toy models(Section 3)
and experiments on LLM activations(Section 4) (Gemma-2-2B, Llama-3.2-1B), the authors show
that reconstruction loss can favor incorrect SAEs and propose a diagnostic metric, the nth decoder
projection score (s dec n ) defined in Equation 5, for selecting appropriate L0 values.

**Strengths:**

1. Clear identification of a critical issue with sparsity tuning in SAEs.
2. Strong empirical evidence with both toy models and comprehensive testing of
LLMs(Gemma-2-2b and Llama-3.2-1b). Specifically, Section 3.3 shows an incorrect,
feature-mixing SAE achieving a better MSE (2.73) than the perfectly correct ground-
truth SAE (MSE 4.88).
3. The proposed (s dec n ) metric (Section 3.5) is well-motivated and is shown to be a useful
proxy for feature correctness.
4. Insightful analysis showing that reconstruction error can misleadingly favor mixed
features.
5. The authors validate their metric on LLMs by showing that the &quot;elbow&quot; of the s dec n
curve (the optimal L0) aligns with peak performance on downstream k-sparse
probing tasks.

**Weaknesses:**

1. Lacks theoretical grounding, findings are entirely empirical.
2. Toy models assume orthogonal and linearly separable ground-truth features, which may
not represent real LLMs.
3. The metric s dec n requires manual hyperparameter tuning (choice of n, batch size). The
paper&#39;s own attempt at an automatic optimization algorithm (Appendix A.6) is admitted
to be hard and &quot;require a lot of hyper-parameter tuning to work in real LLMs limiting its
utility&quot;.
4. The authors explicitly state in Appendix A.8 that LLM experiments were limited to &quot;only
a few layers&quot; due to computational costs, so generalization to all layers remains
uncertain.

**Questions:**

1. Provide theoretical reasoning or formal analysis for why s dec n tracks monosemanticity.
2. Explore robustness under different loss functions and decoder regularizations.
3. Extend experiments to diverse architectures and deeper layers.

---

> ### Author Response · Authors · 2025-11-20
>
> We thank the reviewer for their thorough reading of our work, and for noting that the problem identified is a “critical issue” in SAE training, that the work provides “strong empirical evidence”, that the metric is “well-motivated and is shown to be a useful proxy”. We respond to the questions raised below:
>
> > Lacks theoretical grounding, findings are entirely empirical.
>
> To improve the theoretical grounding of the paper, we have added a formal proof that low L0 incentivizes feature mixing in Appendix A.5.
>
> > Provide theoretical reasoning or formal analysis for why s dec n tracks monosemanticity.
>
> - We added a proof that the $s_n^{dec}$ metric tracks the amount of feature mixing in SAEs (A.10).
> - We also added a proof that $c_{dec}$, now the main metric, also tracks the amount of feature mixing in SAEs (A.6)
>
> > Toy models assume orthogonal and linearly separable ground-truth features, which may
> not represent real LLMs.
>
> We use the simplest possible toy models that should be trivial for an SAE to solve to strengthen the argument that the SAE mixes underlying features together when L0 is too low. If the SAE behaves this way even on trivial problems with no superposition noise at all, we should not expect them to be successful on more challenging tasks.
>
> That being said, we have added further toy model results with superposition noise (A.3.3 and A.4.1), verifying that SAEs still mix together correlated features due to low L0 when there is superposition noise.
>
> > Extend experiments to diverse architectures and deeper layers.
>
> We have extended the analysis to layer 20 of Gemma-2-2b (A.12), so there is now coverage of early, middle, and late layers of the model.

---

### Official Review · Reviewer_Z23o · 2025-11-07

**Soundness:** 1
**Presentation:** 2
**Contribution:** 1
**Rating:** 2
**Confidence:** 4

**Summary:**

The paper investigates the consequences of setting too low an L0 value when training Sparse AutoEncoders (SAEs). Toy experiments show evidence of feature hedging when L0 is set too low, and also indicate that "sparsity versus reconstruction" analyses in prior work can lead to choosing artificially low L0 values (and associated hedging). The authors propose a heuristic method for estimating better L0 values, and conduct several experiments that aim to validate this method.

**Strengths:**

The paper indicates an important issue (feature hedging) that can arise from poor hyperparameter (L0) selection, which is relevant for practitioners to keep in mind when training SAEs. Additionally, the demonstration with toy models that popular "sparsity versus reconstruction" analyses can yield to poor L0 selection may be helpful to the SAE research community.

**Weaknesses:**

It has been a well-established fact since the earliest days of machine learning that poor hyperparameter selection leads to underperforming models. It is not clear to me that the paper makes any contribution beyond showing that this is also true of the L0 hyperparameter when training SAEs.

One potential contribution of the work would be in its proposal of "metric" $s_n^{dec}$ that might be useful in estimating L0 values. However, it is not clear what $s_n^{dec}$ it is intended to measure, nor its intended utility.
- Charitably, such a metric that estimates L0 values might be useful in reducing time and resource costs associated with SAE hyperparameter search. However, for $s_n^{dec}$ in particular, this is not possible, as computing it requires training SAEs first.
- Sec 4 does contain experiments computing $s_n^{dec}$ in the context of BatchTopK SAEs trained on two LLMs, comparing $s_n^{dec}$ and sparse probing scores across L0 values, finding that the two often peak in similar layers. However, it is not clear that analyzing $s_n^{dec}$ has any utility (in terms of theoretical understanding, practical SAE training, or any other consideration) in this setting -- if we already know sparse probing scores, what is the additional benefit of computing $s_n^{dec}$?

Finally, the paper is very light on citation and discussion of closely related prior works, such as:
- [1, 2] study feature splitting, a closely related phenomenon to hedging (as observed in this work for low-L0 SAEs).
- [3] introduces a metric and activation function to approximate the theoretically optimal L2 norm of SAE latent vectors.
- [4] studies feature hedging, as observed in this work -- while the paper cites [4] several times, the novelty of this work relative to [4] is not specified. (Optimistically, the distinction would be that this work focuses on hedging that arises from low L0 values, whereas [4] focuses on hedging due to small dictionary sizes -- however, the effect of L0 on hedging was also shown in [4] (see, e.g., sec 4.1 of [4]).)

[1] Bricken, T., Templeton, A., Batson, J., Chen, B., Jermyn, A., Conerly, T., Turner, N., Anil, C., Denison, C., Askell, A., et al. Towards monosemanticity: Decomposing language models with dictionary learning. Transformer Circuits Thread, 2, 2023. URL: https://transformer-circuits.pub/2023/monosemantic-features#phenomenology-feature-splitting
[2] Templeton, et al., "Scaling Monosemanticity: Extracting Interpretable Features from Claude 3 Sonnet", Transformer Circuits Thread, 2024. URL: https://transformer-circuits.pub/2024/scaling-monosemanticity/#feature-survey-neighborhoods
[3] Lee, S., Davies, A., Canby, M. E., & Hockenmaier, J. (2025). Evaluating and Designing Sparse Autoencoders by Approximating Quasi-Orthogonality. Second Conference on Language Modeling. URL: https://openreview.net/forum?id=XhdNFeMclS
[4] Chanin, D., Dulka, T., & Garriga-Alonso, A. (2025). Feature Hedging: Correlated Features Break Narrow Sparse Autoencoders. arXiv preprint arXiv:2505.11756. URL: https://arxiv.org/abs/2505.11756

**Questions:**

Beyond those raised in the Weaknesses section, a few remaining questions and concerns are listed by section below.

Sec 3.1-3.3:
- Why use fewer "true features" than dimensions ($g < d$) in toy settings, when one of the primary motivations of modern SAEs is that embedding vectors are expected to leverage superposition to encode (potentially exponentially) more features than embedding dimensions ($g >> d$)?
    - If experiments are repeated with $g >> d$, how similar do the results look?

Sec 3.5:
- What does it mean to "sort these values [of $\mathbf{z}$] in descending order to get $\mathbf{z}_\downarrow$"? What is "descending order" (later referred to as "ranking") in this case, and why is it necessary to sort $\mathbf{z}$ according to this ranking?
- Is there any theoretical justification supporting the proposed $s_n^{dec}$ (as defined in eqn 5) as an estimator of the ideal L0 value? (I have the same concern regarding $n = h / 2$.) From the provided explanation, it seems that the authors might have simply experimented across arbitrary L0 thresholds until finding a formula that happened to perform well in toy experiments. However, if this interpretation is correct, then it is not clear whether it is simply due to (unintentional) p-hacking -- i.e., running enough arbitrary experiments that eventually a seemingly-nontrivial pattern appears by chance.

---

> ### Author Response · Authors · 2025-11-20
>
> We thank the reviewer for reading our work and providing detailed feedback. We respond to the questions and concerns raised below:
>
> > poor hyperparameter selection leads to underperforming models. It is not clear … beyond showing that this is also true of the L0 hyperparameter.
>
> We clarify that L0 is not currently treated as just a normal hyperparameter to be optimized in SAE literature. For example, the largest release of open-source SAEs, Gemma Scope by DeepMind [1], released 5 different SAEs per layer with different L0 values. If L0 is a standard hyperparameter that everyone in the field knows should be optimized, it would be very strange to release 5x the SAEs just to cover different L0s. Furthermore, all SAE architecture work we know of evaluates sparsity vs reconstruction as Pareto frontiers, implying all L0 points on the curve are equally valid.
>
> Our work challenges this consensus. We demonstrate that a 'correct' L0 exists for feature recovery, and deviations from it lead to specific failures (hedging/mixing) rather than just a neutral trade-off. The fact that this implies L0 should be treated as a standard hyperparameter is exactly the contribution of our paper: a shift from the current paradigm.
>
> >  if we already know sparse probing scores, … why compute $s_n^{dec}$
>
> If there is already a downstream metric the user cares about they should use that metric as validation. However, we do not always have such a metric. Some of the most exciting uses of SAEs in interpretability come from understanding models trained on novel scientific domains like protein folding, drug discovery, DNA sequences, etc… where validation tasks do not exist.
>
> > Why use fewer "true features" than dimensions ($g < d$) in toy settings?
>
> We chose the simplest setting ($g < d$) to demonstrate that low L0 causes feature mixing even in the absence of superposition noise, as this is a 'best case' scenario for SAEs.
>
> Regardless, we have added experiments with superposition noise ($g > d$) in A.3.3 and A.4.1. These results verify our conclusions are still correct in the presence of superposition.
>
> >  It is not clear what $s_n^{dec}$ measures. Is there theoretical justification supporting $s_n^{dec}$? (...same regarding $n = h / 2$.)
>
> Based on your (and others) feedback, we now use decoder pairwise cosine similarity, $c_{dec}$, as the primary metric; previously it was our “alternative metric”. This metric has the following advantages:
>
> - This metric is parameter-free, so there can be no confusion around how to pick hyperparameters.
>  - It is easier to understand why $c_{dec}$ increases with more mixing of features into SAE latents intuitively and theoretically.
>
> In addition, we have added the following theoretical proofs to the paper to make this more robust:
>
> - Proof that low L0 causes feature mixing in a small SAE (A.5)
> - Proof that $c_{dec}$, now highlighted in the main body, tracks feature mixing and thus L0 (A.6)
> - Proof that $s_n^{dec}$ tracks feature mixing, and thus L0, including the justification for $h/2$ as a max value of $n$ (A.10).
>
> The intuition behind using $s_n^{dec}$ and $c_{dec}$ is as follows: we want to track how much correlated features get mixed (hedged) into SAE latents. Anything that measures how much the SAE latents mix together correlated features will work to find the optimal L0.
>
> For $s_n^{dec}$, the intuition is that projecting the decoder against training inputs should result in a narrow gaussian near 0 for non-activating inputs. The more the decoder mixes correlated features, the wider this projection will be. $n = h/2$ is the median value, which for a gaussian is also the mean, so any value of roughly $n < h/2$ works. See A.10 for a formal proof. Furthermore, almost any $n < h/2$ results in roughly the same elbow point, as we show in A.15.
>
> > What does it mean to "sort these values"? What is "descending order" … why is it necessary to sort?
>
> $s_n^{dec}$ corresponds to the $n/h$-th percentile, e.g., when $n=h/2$, $s_n^{dec}$ corresponds to the 50-th percentile or median value (and thus the mean assuming a gaussian distribution). To find percentiles requires sorting. “Descending order” means sorting from largest to smallest. We have rephrased this in the paper, and added demonstration Python code (A.17).
>
> > the effect of L0 on hedging was also shown in [4] (see, e.g., sec 4.1 of [4]).
>
> This figure is showing the change in hedging when adding latents to an SAE (width-based hedging) at fixed L0, not the effect on existing latents when L0 is varied. In Section 4 of their paper, they explain hedging is measured by increasing the width of the SAE and measuring the effect of the added width.
>
> > … discussion of prior works
>
> We have expanded the related works section and added citations.
>
> ### References
> [1] Lieberum, Tom, et al. "Gemma scope: Open sparse autoencoders everywhere all at once on gemma 2." arXiv preprint arXiv:2408.05147 (2024).

---

### Author Response · Authors · 2025-11-20

We thank the reviewers for their engagement with our work. We highlight that reviewers found our work “provides convincing evidence”, that picking L0 “is a very important problem”, that our analysis is “well-motivated” with “strong empirical evidence”.

Based on the feedback from reviewers, we have made the following improvements to the paper. We have tried to address all concerns raised, but are open to further suggestions for improvement.

## Metric
As suggested by reviewer concerns, we swapped the roles of the $s_n^{dec}$ (previously the main metric) with decoder pairwise cosine similarity $c_{dec}$ (previously the alternative metric). So, $c_{dec}$ is now highlighted more prominently as the main metric in the main body of the paper. This has the following advantages:

- $c_{dec}$ is parameter-free, so there can be no confusion about how to set the hyperparameters.
- $c_{dec}$ is easier to understand. In particular, it is easier to understand why this metric increases if there is more mixing of features into SAE latents, both intuitively and theoretically.

## Theory
Several reviewers mentioned they wanted more theoretical justification, so we added the following:

- Proof that low L0 causes feature mixing in a simple SAE (A.5)
- Proof that feature mixing causes larger values of $c_{dec}$ (A.6)
- Proof that feature mixing causes larger values of $s_n^{dec}$ (A.10)

## Experiments
Several reviewers asked for further experiments, so we ran these and added them to the paper:

- Experiments showing that the degree of feature mixing is determined by amount of feature correlation (A.3.1)
- Experiments showing that the degree of mixing increases as L0 decreases below the true L0 (A.3.2)
- Experiments with superposition noise (A.3.3, A.4.1)
- More LLM experiments (A.12)

## Misc
We also made the following changes based on reviewer feedback:

- Improved writing and presentation clarity
- Added Python pseudo-code for metrics (A.17)
- Added all references reviewers requested to Related Work (S.5)

We believe this addresses the concerns raised by reviewers, and makes the paper much stronger as a result.

---

*update: 2025-11-30*

We thank all reviewers for their feedback during the review process. We specifically wish to highlight Reviewer cz9D’s final comment, with which we strongly agree:

> I want to begin my comment by disagreeing with some of the statements from reviewer Z23o. The comment that 'It has been a well-established fact since the earliest days of machine learning that poor hyperparameter selection leads to underperforming models. It is not clear to me that the paper makes any contribution beyond showing that this is also true of the L0 hyperparameter when training SAEs.' **seems to either ignore all the SAE literature up until today or to ignore it.** It also seems to ignore the multitude of literature that exists in machine learning research indicating how to better chose hyperparameters.

We hope this helps clarify that the prevailing paradigm in SAE research (e.g., as demonstrated via standard sparsity-reconstruction trade-off plots) treats L0 as a matter of user preference with no correct value. Our work challenges this consensus by demonstrating that a "correct" L0 exists and that deviations from it corrupt the SAE dictionary. We believe the discussion between reviewers has clarified that establishing L0 as a critical hyperparameter, rather than as just a preference, is a significant contribution to the field.

---

### Meta-Review · Area_Chair_gbfY · 2026-01-10

**Summary:**

This paper presents a series of well-done, carefully controlled/synthetic experiments on sparse autoencoders (SAEs), showing that (a) an incorrect choice of sparsity (L0) parameter leads to poor disentanglement behavior, and (b) proposing a data-distribution-dependent metric that can help inform the correct choice of L0.

The paper received widely divergent reviews. Primarily, there was disagreement among the reviewers on the significance/impact of the work. One reviewer pointed out that the only (and somewhat obvious) contribution is that hyperparameter selection in SAEs is essential; another reviewer disagreed, saying that hyperparameter selection is an important problem in SAEs and in general.

There was also a concern about the utility of the proposed metric, and whether there really is any benefit of using it in practice. The authors responded by replacing that metric with another metric.

Other concerns included insufficient discussion and comparisons relative to the existing literature, lack of theory, limited experiments, and lack of robustness results.

While many of the latter concerns were largely addressed in the rebuttal, I personally side with the negative reviews. At the core, the experiments are on the L0-parameter selection problem for a (very) synthetic set of dictionary learning tasks. Unclear whether the synthetic data distribution mirrors real world settings. Choosing the right parameters for dictionary learning dates back at least to the mid 2000s (well before the problem got rebranded/recast as SAE learning) --- and there is barely any discussion of that entire body of literature here. Many insights (e.g. "if L0 is high then latents are learned, but it it is low then all latents are affected") are folklore consequences of compressed sensing theory. The theoretical results from A.5 and A.6 also are somewhat rudimentary compared to such results. The c_dec metric also seems somewhat heuristic. (Why consider summation of absolute pairwise correlations? Why not the max, median or some other statistic?)

Overall, while this is nice work, I would not support acceptance to ICLR.

**Reviewer Concerns:**

I think the concerns surrounding clarity of writing and additional experiments were satisfactorily addressed, but the concerns around significance of the problem as well as the well-foundedness of the proposed metric continue to persist.

**Reviewer Scores:**

The initial scores were 2, 4, 6, and 8. I think the 6 and 8 reviewers would have kept their scores (the 8 reviewer said as much in a comment after the reviews were posted).

The 4 reviewer raised several important concerns. I feel that the responses could have been better. I don't think they would have changed their score.

The 2 reviewer raised the concern about significance, and also questioned the utility of the metric. I don't think they would have changed their score.

---

### Decision · Program_Chairs · 2026-01-26

Reject